# A non-spatial account of place and grid cells based on clustering models of concept learning

Robert M. Mok [1]* & Bradley C. Love [1,2]*

One view is that conceptual knowledge is organized using the circuitry in the medial temporal lobe (MTL) that supports spatial processing and navigation. In contrast, we find that a domain-general learning algorithm explains key findings in both spatial and conceptual domains. When the clustering model is applied to spatial navigation tasks, so-called place and grid cell-like representations emerge because of the relatively uniform distribution of possible inputs in these tasks. The same mechanism applied to conceptual tasks, where the overall space can be higher-dimensional and sampling sparser, leading to representations more aligned with human conceptual knowledge. Although the types of memory supported by the MTL are superficially dissimilar, the information processing steps appear shared. Our account suggests that the MTL uses a general-purpose algorithm to learn and organize context-relevant information in a useful format, rather than relying on navigation-specific neural circuitry.

[1] Department of Experimental Psychology, University College London, 26 Bedford Way, London WC1H 0AP, UK. [2] The Alan Turing Institute, London, UK.
*email: robert.mok@ucl.ac.uk; b.love@ucl.ac.uk

Concepts organize experiences to enable generalization and inference. For example, a traveler encountering an unfamiliar bird species would reasonably infer the bird was born from an egg. One longstanding question is the basis for people's abstract conceptual knowledge. One intuitive idea is that concepts ground in a more basic and concrete substrate, such as sensory-motor experience[1]. For example, abstract concepts such as time may be represented in terms of experience of space[2]. Relatedly, conceptual knowledge may be organized using circuitry in the medial temporal lobe (MTL) that supports navigation[3].

This view is supported by recent studies that find the brain's responses to conceptual tasks parallel those previously found in spatial tasks. Place cells in the hippocampus[4] typically have single firing fields at circumscribed locations in a spatial environment, and grid cells in the medial entorhinal cortex (mEC)[5–7] display multiple regularly-spaced firing fields arranged in a hexagonal pattern covering the environment. These spatially-tuned cells in the MTL are thought to implement a spatial cognitive map for navigation[8–11], and recent work suggests these cells also represent conceptual[12] and task spaces[13]. One key question is whether the same brain systems and computations support concept learning, memory, and spatial navigation.

One neglected possibility is that the relation between spatial and conceptual representations has been framed backwards. Perhaps, rather than concepts grounding in the machinery of navigation, spatial concepts are a limiting case of a single, more general, learning system. Such a learning system would be tasked with learning all relevant concepts, including those tied to physical space (also see refs. [14,15], and Discussion). This general learning system would support learning concepts, which are typically clumpy in that they consist of clusters of interrelated features in a high-dimensional space[16]. For example, animals that fly tend to be small and have wings (see Fig. 1a). Not all possible combinations of features are relevant and represented. In contrast, many spatial tasks[5] and their conceptual analogs[12] typically involve a uniform and exhaustive sampling of all possible combinations within a low (two-) dimensional space corresponding to locations in an environment (see Fig. 1c, d).

We evaluate whether a domain general account is plausible by applying successful models of human concept learning to spatial contexts. In concept learning studies, these models learn to represent experience in terms of conceptual clusters, which are not uniformly distributed[17]. For example, in a simple case with two clearly separable and internally coherent sets of objects, a clustering model would use one cluster to represent each concept, each of which would be centered amidst its members in representational space (e.g. Fig. 1a).

When the model is presented with a novel item, the closest cluster in representational space is activated, which signals the category membership of the item. An error-monitoring signal gauges how well an item matches this closest cluster in representational space. In these models, only the closest cluster maintains non-zero activation (winner-takes-all), so an error-monitoring signal (entropy term) 'monitors' activation of all existing clusters which indicates how close or far away the current location was from any cluster, acting as a cluster match (or non-match signal)[17,18].

These clustering representations successfully capture patterns of activity in the MTL[19,20] and are in accord with the notion that the human hippocampus contains concept cells in which individual cells respond to a specific concept, much like how a cluster in a possibly high-dimensional space can encode a concept[21]. Analogously, place cells respond to a location in a particular two-dimensional spatial context. It is important to note that clusters are abstract entities in the model, and there need not be a one-to-one mapping to single concept or place cell (e.g. a cluster can be represented by a group of place cells with similar tuning (c.f. refs. [22])—a functional mapping of multiple place cells to one cluster, and the place cell population to the whole cluster representation; Fig. 2c). Furthermore, clustering models explain how individual episodes give rise to conceptual knowledge over the course of learning[23], consistent with both the hippocampus's importance in memory[24,25]. We evaluate whether the same mechanisms also offer a general understanding of place and grid cells, and their relationship to concepts.

To facilitate this evaluation, we simplified the clustering models to only include aspects necessary for this contribution. Clustering models that capture behavior on a trial-by-trial basis typically recruit a new cluster in response to a surprising error. These models also learn attention weights that accentuate task-relevant stimulus dimensions and associate clusters with behavioral responses (e.g., respond "bird"). Without loss of generality, we

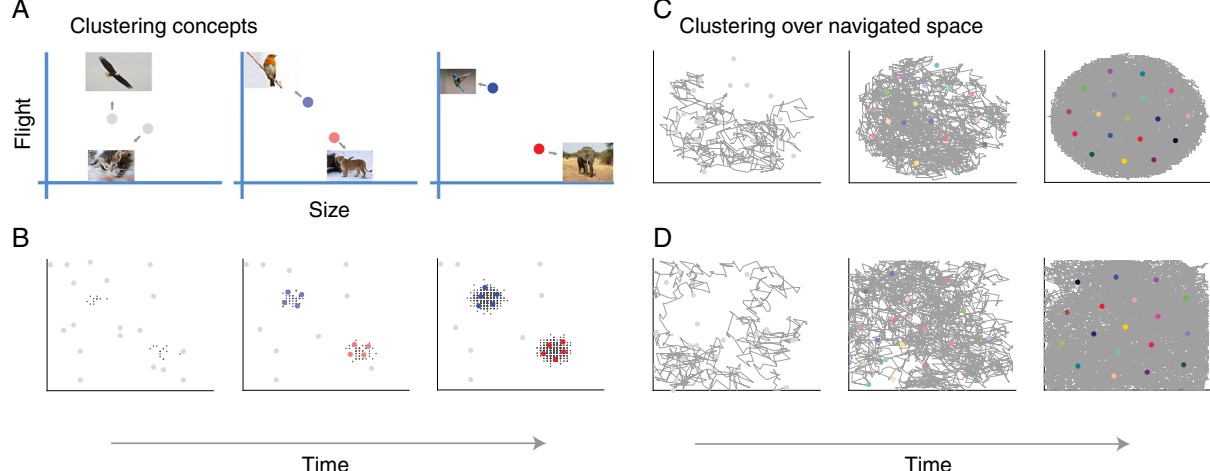

**Fig. 1 Cluster learning applied to conceptual and spatial examples. a** The most similar cluster moves (i.e., adjusts its tuning) toward its newest member and becomes associated with a response (blue for bird, red for mammal). **b** Out of a pool of many randomly tuned clusters, a subset comes to represent the two concepts over learning. **c, d** The same learning system applied to an agent locomoting in a circular or a square environment gives rise to a hexagonal cluster organization. How the stimulus space is sampled affects how clusters are distributed in the representational space.

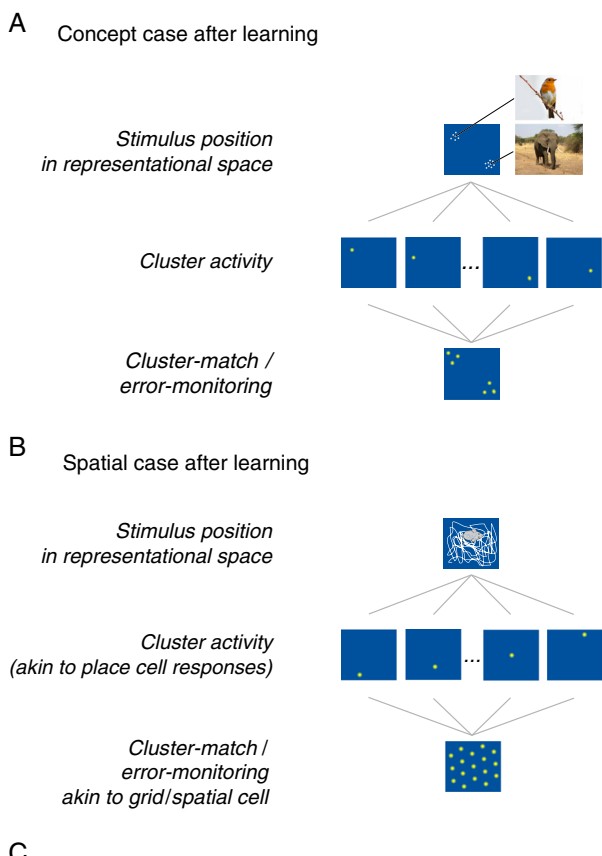

A   Concept case after learning

*Stimulus position in representational space*

*Cluster activity*

*Cluster-match / error-monitoring*

B   Spatial case after learning

*Stimulus position in representational space*

*Cluster activity (akin to place cell responses)*

*Cluster-match / error-monitoring akin to grid/spatial cell*

C   Abstract cluster representations are encoded by multiple neurons

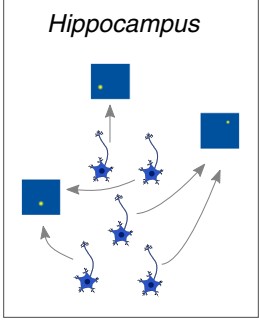

*Hippocampus*

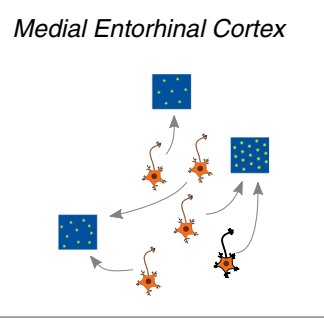

*Medial Entorhinal Cortex*

**Fig. 2 Cluster representations after learning for conceptual and spatial tasks. a** Clusters clump into two groups. Thus, novel bird and mammal stimuli will strongly activate one or the other grouping, which does not lead to a grid response across possible stimuli. **b** In contrast, for the spatial case, clusters form a hexagonal grid which leads to a grid-like response across possible stimuli when cluster activity is monitored. **c** Clusters determine the receptive fields for a population of place or concept cells, and the cluster-monitoring/error-monitoring mechanism (grid or spatial cells) reflect the distribution of the clusters. Abstract cluster representations are instantiated by multiple cells in the hippocampus and medial entorhinal cortex (mEC) with similar firing fields to represent the same location (or concept) in the case of hippocampal cells or cluster match in the case of mEC cells.

simplified the models by pre-seeding with a fixed number of clusters and limiting learning to updating cluster positions. In particular, the cluster most similar to the current stimulus updated its position in representational space to be closer (more similar) to its newest member (see Methods for full details), much like cluster updating in Kohonen learning maps[26] and k-means clustering[27].

## Results

**A common learning mechanism for space and concepts**. As shown in Fig. 1a, the model when applied to categorizing animals as birds or mammals learns to segregate the items into two groupings. These clusters can be seen as concept cells, akin to place cells (Fig. 2a, b). Notice that the items (i.e., experiences) and the clusters only cover a select portion of the stimulus space. For example, no animal exists that is as massive as an elephant and can fly. Clustering solutions capture the structure of the environment, which enables generalization to novel cases.

In contrast, the same model applied to an agent exploring a typical laboratory environment leads to clusters that uniformly cover the entire representational space in a hexagonal pattern (see Fig. 1c, d). In the spatial case, there is no salient structure present in the input to the model, which results in clusters covering the representational space, much like how a bunch of tennis balls dropped into a square box will self-organize into a grid-like lattice according to the mathematics of packing[28–33]. In the spatial case, the clusters function in a similar way to a population of place cells that code for (i.e., discriminate) locations.

In our account, grid-like responses arise from monitoring the match (inverse error) of the clustering solution (Fig. 2a, b). In unsupervised learning, error or uncertainty is simply the inverse of how similar an item is to the best matching cluster. Notice that matching clusters in the spatial case should display a hexagonal pattern because of the hexagonal clustering pattern in representational space, resulting in canonical grid-like receptive fields (see Fig. 2b). In the conceptual case, we predict that typical grid cell firing patterns should not be observed because the clusters (i.e., place cells) do not form a hexagon pattern (Fig. 2a) in representational space. One might object that our account is inconsistent with conceptual learning brain imaging studies that find grid-like response patterns[12]. However, these studies are consistent with the model because they follow the design principles of typical spatial studies—all feature combinations within a 2-dimensional stimulus space are sampled, which would lead to a hexagonal clustering solution (Fig. 2b).

**Clustering solutions match grid patterns in mEC**. To relate our account to typical spatial studies, we simulated an agent moving through its environment as in a free-foraging rodent experiment. As expected, learning led to clusters forming a hexagonal pattern (see examples in Fig. 3a, b, left). To assess this quantitatively, we computed the spatial autocorrelograms of the cluster activation maps (Fig. 3a, b, right) to obtain the grid score, which reflects the degree six-fold hexagonal symmetry in the cluster activation pattern across trials[5] (see Methods). We computed grid scores for each time bin during learning and found that grid scores tended to increase over learning in both the square (see Fig. 3c for examples and Fig. S1 for all conditions; mean slope: 0.0044, bootstrap CIs: [0.0040, 0.0048]) and circular environment (mean slope: 0.0042, bootstrap CIs [0.0038, 0.0046]; see Tables S1 and S2).

Following learning, we evaluated the gridness of the clustering solution (see examples in Fig. 3a, b and Figs. S2 and S3). A substantial proportion of simulations satisfied the criterion for grid-like organization, with 45.3% in the square and 38.6% in the circular environment, which closely match the proportions in empirical results (45% and 38%, respectively; see Supplementary Note 1)[34,35]. The average grid score in both the square environment (mean: 0.277, bootstrap CIs [0.273, 0.0280]) and circular environment (mean: 0.313, bootstrap CIs [0.309, 0.318]) were greater than zero; see Tables S3 and S4.

**Cluster representations are shaped by environmental geometry**. According to the clustering account, grid-like responses should

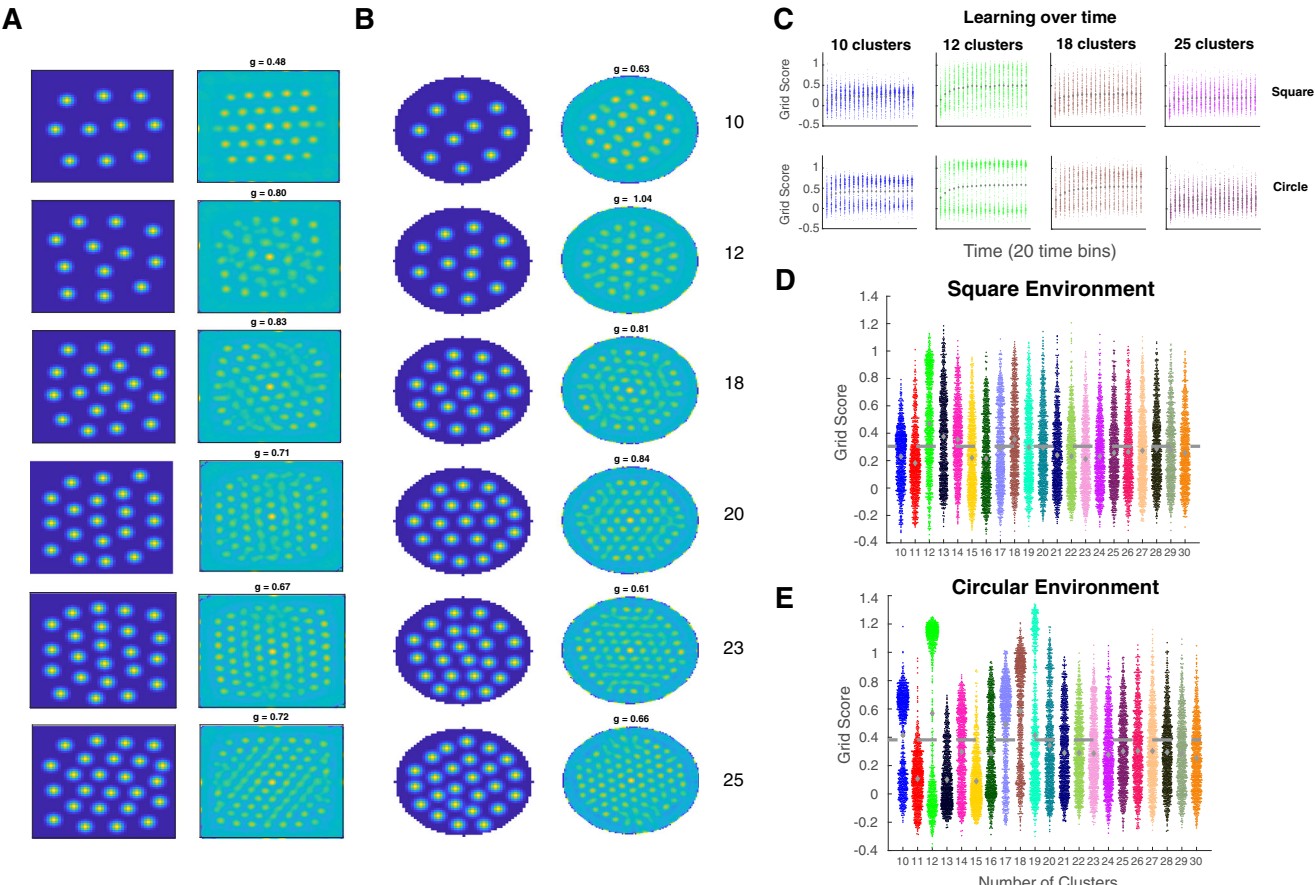

**Fig. 3 Clustering leads to activation maps similar to spatial cells in medial entorhinal cortex. a, b** Examples of activation maps with grid patterns in a square environment (A-left) and their corresponding spatial autocorrelograms (A-right), and activation maps in a circular environment (B-left), and spatial autocorrelograms (B-right). **c** Examples showing grid scores increasing over learning in the square (top) and circle (bottom). **d, e** Univariate scatterplots showing grid scores for simulations in the square (**d**) and circle (**e**). Dashed line represents the most conservative threshold for a 'grid cell'.

only arise under very specific conditions in which the environment is fairly uniform. The imposition of any structure, including changes to the overall geometry of the environment, should affect the clustering in a manner that makes it less grid-like.

Related, Krupic et al.[36] identified grid cells in rodent mEC in a square box, then placed the animals in a trapezoid environment. They found that activity maps of grid cells became less grid-like in the trapezoid and that the decline was greatest for responses elicited on the narrow side of the trapezoid. To simulate this experiment, the model was first trained in a square and then transferred to a trapezoid environment (see Fig. 4a–d for an example and Fig. S4 for more examples). As in the empirical studies, the model's overall grid scores declined in the trapezoid environment (Fig. 4e; trapezoid mean grid score: 0.058, bootstrap CIs [0.054, 0.061]; Fig. 4f; square minus trapezoid mean: 0.219, bootstrap CIs [0.214, 0.224]) and the grid scores were higher on the wide than on the narrow side of the trapezoid (Fig. 4g; wide minus narrow mean: 0.133, bootstrap CIs [0.127, 0.139]; see Tables S5–S7).

## Discussion

Previous work has explained a wide array of learning and memory phenomena in terms of clustering computations supported by the MTL[23]. Here, this same basic account was shown to account for basic spatial navigation phenomenon, including place and grid cell-like response patterns. Specifically, we showed that a learning mechanism that seeks to minimize error in the task-relevant feature space captures conceptual structure in concept learning tasks and spatial structure in two-dimensional navigation contexts, which lead to place and grid cell-like representations. Rather than spatial mechanisms providing a scaffolding for more abstract conceptual knowledge[3,10], the current results suggest that key findings in the spatial literature naturally arise as limiting cases of a more general concept learning mechanism. Whereas concepts can be clumpy, structured, and high dimensional, typical spatial tasks involve exhaustive and uniform sampling of simple two-dimensional environments, which leads to degenerate clustering solutions that pack clusters into a hexagon lattice, giving rise to so-called grid cells (Fig. 3). The clustering account correctly predicted how deviations from these unstructured learning environments should reduce grid-like cell responses (Fig. 4).

Our proposal stands in contrast to other ideas that a dedicated, phylogenetically older spatial navigation system in the MTL supports the newer, higher-level cognitive functions[3,10]. In particular, we suggest there are no intrinsic 'place' or 'grid' cells, but instead a flexible system that will represent the relevant variables at hand, including physical space. Emerging evidence shows that cells in the MTL exhibit mixed-selectivity in that they respond to multiple variables, such as place and grid cells that also code for task-relevant sound frequency[13], routes[37,38], objects and context[39], and time[40], suggesting a flexible code. Clustering is a flexible mechanism and can learn representations in multi-dimensional space, and therefore is a strong candidate mechanism for organizing multi-modal, complex information for

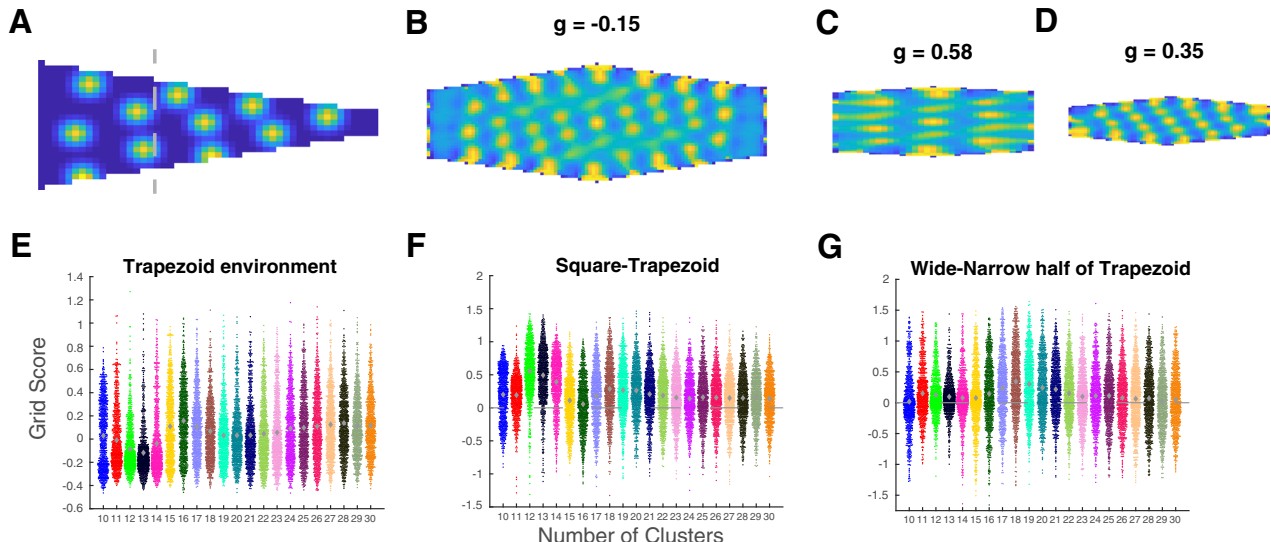

**Fig. 4 The clustering model captures declines in grid responses in trapezoid environments. a–d** Example of distortion in a trapezoid environment. **a** Example of an activity map in a trapezoid environment with 18 clusters. The dotted line demarcates the wide and narrow halves of the trapezoid. **b** Spatial autocorrelogram of the trapezoid. **c** Spatial autocorrelograms of the wide (left) and **d** narrow (right) portion of the trapezoid in **a**. **e** Univariate scatterplot showing grid scores for simulations in the trapezoid enclosure after learning in a square enclosure. **f** The difference (positive) between grid scores in the square and trapezoid. **g** The difference (positive) between grid scores for the wide and narrow halves of the trapezoid.

consolidation of knowledge for memories and concepts. A growing body of work supports the idea that the MTL system is a key part of a general learning system for organizing knowledge into a useful representation, which can be used for effective behavior and for memory consolidation.

Clustering models organize information and represent concepts in feature space to enable the identification, classification, and generalization of novel objects[17,18]. These models can be closely linked to episodic memory[23] and accounts of hippocampal function including relational memory[41], statistical learning[42], and transition statistics (successor representation[14,43–45]), with objects or memories arranged in the form of a cognitive map.

Several recent accounts have proposed different mechanisms for the hippocampal-entorhinal cell circuit in organizing non-spatial information[10,14,15]. One major feature that distinguishes our account is the role of place and grid cells. In our account, the hippocampus plays a central role in organizing information about the current environment or task, and the mEC monitors these hippocampal representations. As such, mEC cells do not play a representational role, but play a role in learning—monitoring error from existing clusters in order to update the cluster representation. Both grid and non-grid spatial cells contribute to this function, and the high gridness of a subset of these cells is a result of the environment or space.

Other accounts hold that grid cells are key representational units in the cognitive map. For example, Stachenfeld and colleagues[14] suggested that place cells encode predictions of future states, and grid cells encode a low-dimensional decomposition of this hippocampal predictive map that may be useful for stabilizing the map and representing sub-goals. In contrast, we suggest that place cells (clusters) are the key representational units which encode locations in representational space and its structure, whereas grid cells monitor place cell activity. Behrens and colleagues[15,46] proposed that the hippocampal-entorhinal circuit learns and represents structural knowledge useful for generalization. This account assumes objects are represented in lateral EC (lEC), structure is represented in mEC, and the hippocampus encodes conjunctions of the two. The learnt structural information in mEC can be used to generalize to different contexts with shared structure. In our account, conceptual knowledge and its structure is represented in the hippocampus, and any generalization to new instances from existing structure is from hippocampal representations (as generalization is performed in clustering models of concept learning). In contrast to the view that hippocampal representations arise from interactions between mEC and lEC, we argue for a central role of prefrontal cortex (representation of the task or relevant features) for shaping hippocampal representations[19], in combination with sensory inputs arriving via entorhinal and perirhinal cortex, and from anterior inferior temporal cortex to prefrontal cortex[47,48] to the hippocampus.

Whereas our account holds that place and grid cells emerge from a general learning system, Bellmund and colleagues suggest that the population code of place and grid cells play a role in mapping the dimensions of cognitive spaces in cognitive tasks, and that spatial navigation could serve as a model system to understand cognitive spaces[10] (also see ref. [3]). Although there are commonalities, their proposal suggests that place and grid cells provide or a 'metric' or distance code for abstract spaces, and that there is a straightforward mapping from neural representations of physical space to abstract space. In our view, when the context involves a significant degree of selective attention to stimulus features or task variables, the representational space can be warped to a different, more effective representation of the context at hand (e.g. reducing dimensionality by attending to the task-relevant dimensions[49]), which does not simply map onto the two-dimensional spatial case.

Our higher-level account provides a general theoretical framework applicable to a large range of tasks, in contrast to lower-level models of place and grid cells which make specific predictions in spatial contexts but have less explanatory power to generalize across contexts. Our model's contribution is providing a general mechanism that could be used across domains. Here, we provided an algorithmic-level model[50] that links across two different computational accounts of task descriptions (spatial and concept tasks), and connects learning mechanisms from concept

learning to spatial representations found at the single-cell level. Specifically, we were able to link the model representations to neural measures reported in the spatial literature, closely matching a number of empirical observations.

Our model showed a similar proportion of grid-like cells in found in mEC. Other models either do not capture the hexagonal code[14] (90 degree grids) or need to build in additional constraints[32] (non-negativity constraint changes the 90 degree grid patterns to 60 degree grid patterns). When they find a large proportion of grid cells, they are *too* good in that all the simulated cells are grid cells[32,51]. Other work have modeled or analyzed mathematical properties of the grid code (e.g. refs. [31,33]), but also do not account for variability in the grid score in mEC cells. Here, we used a simple model from a high-level perspective based on ideas from concept learning and memory and matched the proportion of grid cells with empirical data, suggesting that the constraints of the clustering model matches the constraints the brain uses to build these representations.

Our model also captured the causal relation between place and grid cells. In our account, grid cells play a cluster-match or error-monitoring function where they monitor (connected to and receive input from) place cells, and self-organize over time to produce a hexagonal firing pattern. This is consistent with developmental work[52,53], where place cells appear in baby rats very early in life, and grid cells develop shortly after, as they explore and learn about spatial environments during normal development. Furthermore, inactivation of the hippocampus (with place cells) leads to grid cells in mEC losing the periodicity of their firing fields[54], whereas inactivation of the mEC (with grid cells) only mildly affect hippocampal place fields[55]. Our account provides a different way of thinking about hippocampal-mEC interactions, which makes predictions that can guide future experiments and analyses.

Our account suggests that grid-like responses from the MTL should be the exception, not the rule, when encoding abstract spaces. Outside the typical laboratory study, representational spaces may be high dimensional and not all dimensions or values along dimensions will be equally relevant, nor will all combination of values across dimensions (see Fig. 1a). In support of this characterization, empirical work has shown that grid cells also lose their grid-like properties in more complex environments such as mazes[56].

Our account made several predictions that matched empirical data, where changes in environmental geometry lead to specific changes in the cluster representation. The model also provides further predictions. First, the mapping from place to grid cells within a context should be predictable. An mEC grid or spatial cell is assumed to receive input from multiple place cells in the hippocampus, and that mEC cell should have fields in the same location as the place cells it receives input from (Fig. 2a, b). Therefore, if place cells that represent a certain location are inactivated, the corresponding fields of the mEC cells that monitor those place cells should also disappear. Since an mEC cell may receive inputs from multiple place cells, a strict test would require inactivation of all (or at least a large proportion of) place cells that represent one location (a cluster in the model), predicting all mEC cells should also lose those fields. Future work with large-scale concurrent recordings in multiple brain regions with specific (e.g. optogentic) manipulation may allow these predictions to be tested. One novel prediction of our model is that when error is high early in learning for a particular location, mEC cells should show a low firing rate and that best matching place cells should update their tunings to more strongly respond at that location (i.e., cluster updating). Updating a cluster (or recruiting a new cluster) should result in adjustment to the tuning of neighboring clusters, leading to a cascade of changes across place cells.

When error is low, this signifies a good match between the environment and one's current knowledge (cluster representation) and experience, and little or no update is necessary. Inactivating the mEC should disrupt the error signal, which should disrupt learning in new environments. Recent evidence suggests place and grid cells both move towards goals or rewards[57–59], and there seems to be a greater number of place cells recruited near goal locations[60,61] consistent with more clusters moving towards the goal or more clusters recruited at locations near the goal. Finally, our model predicts that both grid and non-grid spatial cells should perform the same function, in both concept and spatial tasks. There is some evidence in the spatial domain which showed that non-grid spatial cells in mEC contain as much spatial information as grid cells and could serve similar functions[62].

The primary strength of our account, namely that it offers an algorithmic account of spatial and concept learning tasks, serves to highlight the need for complementary lower-level accounts. There are various open questions such as how place cell remapping occurs across contexts and partial remapping effects with disruption to mEC[63,64]. Our hope is that our model can eventually link to lower-level models that incorporate biological details such as spiking neurons and incorporate knowledge from memory research that can explain more empirical findings and provide new insights to these questions. Accounts are needed at multiple levels of analysis. We view our model as intermediary (at the algorithmic level) and aim for it to serve as a bridge between the goal of the computation and its implementation. Our model can serve as a guide for how operations such as cluster updating are physically realized.

Building this integrative bridge between the concept, memory, and spatial literatures allows for findings from one domain to inform the other. For example, task goals and attentional mechanisms in the concept literature have been found to shape hippocampal representations[19,20]. Analogous tasks can readily be constructed to evaluate whether spatial cells support broader information processing functions (cf. ref. [13]) and how general learning algorithms shape their response properties (cf. ref. [14]). Likewise, the concept literature emphasizes the hippocampus's interactions with other brain areas, such as medial prefrontal cortex, to assist in encoding task relevant information[19]. When richer spatial tasks are considered, there is a ready set of candidate mechanisms and neural systems that may offer domain general explanations that link across brain, behavior, and computation.

## Methods

**Simulations**. A simulation run comprised of a learning period with a million trials (training phase) where clusters updated their positions in relation to the agent's position as it explored the environment. After learning, we quantitatively assessed the regularity of the cluster position arrangements (test phase). We ran 1000 simulation runs for each condition (number of clusters).

**Simulation procedure and model specifications**. At the beginning of the learning phase of each simulation run, we set the number of clusters, number of learning trials, the environment (square, circle), the learning rate, and the learning update batch size. The number of clusters were set (ranging from 10 to 30) and were initiated at random locations in the environment. The shape of the environment was defined by a set of points that could be visited by the agent. The square environment was 50 by 50, where each point was a location specified by a value on the x- and y-axis. The circular environment was defined by drawing a circle in Matlab with a radius of 50, and selecting the points within the bounds of the circle. The starting position and movement trajectory of the agent was then determined as a random walk over one million trials. The agent started at a random position and steps in the horizontal and vertical axes were computed separately. On each trial, the agent could go up, down, or stay on the vertical axis, and left, right, or stay on the horizontal axis. The step was sampled from [−4, −2, −1, −1, 0, 1, 1, 2, 4], where negative values are steps to the left, positive steps are steps to the right, and zero means stay. Movement on the vertical dimension was determined in the same way, but negative values were upward steps and positive values were downwards

steps. If the generated step brought the agent out of the environment, the step was cancelled and a new step was generated as above.

We considered a simple winner-take-all network in which only the cluster at position $\mathbf{pos_i}$ closest to stimulus $\mathbf{x}$ (agent's location) had a non-zero activation. Bold type is reserved for vectors. The distance between $\mathbf{pos_i}$ and $\mathbf{x}$ is defined as:

$$\mathbf{dist_i} = \|\mathbf{pos_i} - \mathbf{x}\| \qquad (1)$$

In the Kohonen learning rule, cluster $i$ updates its position $\mathbf{pos_i}$ to move toward stimulus $\mathbf{x}$ according to:

$$\Delta\mathbf{pos_i} = \eta_t \cdot (\mathbf{x} - \mathbf{pos_i}), \qquad (2)$$

where $\eta_t$ is the learning rate at time $t$. In the present simulations, we used batch updating to increase numerical stability in which 200 updates were performed simultaneously. The learning rate $\eta$ for batch time $t$ followed an annealing schedule:

$$\eta_t = \frac{\eta_0}{1 + \rho \cdot t}, \qquad (3)$$

where $\eta_0$ is the initial learning rate set to 0.25 and $\rho$ is the annealing rate set to 0.02.

**Assessing regularity of cluster positions**. To assess whether cluster positions formed a regular hexagonal structure with learning in a comparable manner way to grid cells found in the medial entorhinal cortex (mEC), we followed the method of Hafting et al.[5] and Perez-Escobar et al.[35].

In Hafting et al.[5], rodents traversed circular and square environments whilst they recorded electrophysiological signals from mEC neurons. They found cells that displayed multiple firing fields and resembled a grid of regularly tessellating triangles spanning the recorded environment. To assess this regularity quantitatively, they computed the spatial autocorrelogram of the firing rate map. If the fields were arranged in a regular grid, the center peak of the autocorrelogram should be surrounded by six equidistance peaks, forming a regular hexagon. The spatial autocorrelogram was computed as follows. With $\lambda_1(x, y)$ denoting the cluster activation at location $(x, y)$, the autocorrelation with spatial lags of $\tau_x$ and $\tau_y$ was estimated as:

$$r(\tau_x, \tau_y) = \frac{n\sum\lambda_1(x,y)\lambda_2(x - \tau_x, y - \tau_y) - \sum\lambda_1(x,y)\sum\lambda_2(x - \tau_x, y - \tau_y)}{\sqrt{n\sum\lambda_1(x,y)^2 - (\sum\lambda_1(x,y))^2} \times \sqrt{n\sum\lambda_2(x - \tau_x, y - \tau_y)^2 - (\sum\lambda_2(x - \tau_x, y - \tau_y))^2}}, \qquad (4)$$

where $r(\tau_x, \tau_y)$ is the autocorrelation between bins offset of $\tau_x$ and $\tau_y$, $\lambda_1(x, y)$ and $\lambda_2(x, y)$ are equivalent for an autocorrelation indicates the average firing rate of the cell in each location $(x, y)$, and $n$ is the number of spatial bins over which the estimation was made.

To quantify the degree of this regularity, a 'grid score' is commonly used[35] by computing the correlation between the center region of the spatial autocorrelogram (a masked region including the six surrounding peaks but excluding the centre peak) and a 60° and 120° rotated version (to assess the six-fold hexagonal symmetry) minus the correlation between the spatial autocorrelograms and a 30°, 90°, and 150° rotated version (where there should be a low correlation):

$$\frac{(r_{60°} + r_{120°})}{2} - \frac{(r_{30°} + r_{90°} + r_{150°})}{3} \qquad (5)$$

To assess the regularity of the cluster positions in a given environment in the current study and compare our results with empirical findings, we followed the method described above. We first computed activation maps to emulate firing rate maps in empirical neuronal recordings, and computed the spatial autocorrelogram to obtain the grid score.

**Assessing change in gridness during and after learning**. To characterize how cluster positions changed over time in the learning phase, activation maps were computed over trials during learning in a set of 200 simulation runs. Trials were binned into 20 equally spaced time bins with 50,000 trials in each time bin. We assumed that the activation strength of the winning cluster was a Gaussian function of distance from the agent:

$$act_i = \frac{1}{\sqrt{2\pi^2}} e^{-\frac{1}{2}dist_i^2}, \qquad (6)$$

where $act_i$ is cluster $i$'s activation strength. To compute activation maps for each time bin, activations were computed at each location and normalized by the number of visits by the agent (as done in empirical studies) to create a normalized activation map. The maps were smoothed (Gaussian kernel, SD = 1), spatial autocorrelograms were computed, and grid scores were computed for each time bin. As the clusters moved continuously over time (not defined by the time bins), activation maps changed over each time bin.

To test whether gridness increased over time, we used a linear model to estimate the slope (beta value) of the grid score of activation maps over each time bin (20 bins) for each simulation run during the learning phase. For each condition (number of clusters), we estimated the slope for 200 simulation runs, giving

200 beta values. We computed the mean and bootstrapped 95% confidence intervals (CIs) over all conditions and simulation runs to test if the grid score increased over time. We also computed the mean and bootstrapped 95% CIs over the 200 beta values for each condition.

To assess gridness at the end of learning, a new movement trajectory was generated with 100,000 trials and cluster positions were fixed. Grid score after learning was assessed for all 1000 simulation runs. The activations and normalized activation map were computed over all test trials, the activation map was smoothed (Gaussian kernel, SD = 1) and the spatial autocorrelogram of the activation map was computed following Hafting et al.[5], except firing rates were replaced with normalized cluster activation values at each location. Grid scores were then computed based on the spatial autocorrelograms using Eq. (5). We computed the mean grid scores and bootstrap 95% CIs over all conditions and simulation runs. We also computed the mean and bootstrap 95% CIs over each condition.

**Classification and percentage of grid cell-like maps**. To assess whether activation maps showed a regular hexagonal pattern that would be classified as a 'grid cell' according to criteria set in empirical studies, and to compare the percentage of grid-like activation maps from our clustering model to the percentage of grid cells found in the mEC, we used a shuffling procedure to find the statistical threshold of the grid score that passes the criterion for a 'grid cell' described in Wills et al.[52].

The procedure was performed on spatial autocorrelograms of the activation maps produced on the test phase, where cluster positions were fixed. Since cluster activations were generated in relation to the agent's location during movement, they were temporally correlated. Therefore, to break the location-activation relationship, the vector of activations were randomly shuffled in time, and we ensured that each location was at least 20 trials from its original position. The activation map was smoothed (Gaussian kernel, SD = 1) then the grid score was computed. For each condition, this shuffling procedure was performed 500 times on each simulation run (on a subset of 200 simulations). The threshold was defined as the 95th percentile of the 500 shuffled grid scores, giving 200 threshold values (from each simulation run) per condition (number of clusters). The highest threshold value (most conservative) was used as the threshold for each condition. In the figure in the main text (Fig. 3d, e), the thresholds plotted are the highest (most conservative) thresholds across all conditions in that particular environment.

For each condition, we computed the percentage of activation maps that exceeded the shuffled grid score threshold. We computed the percentage of 'grid cells' for each condition (number of clusters) separately and then computed the mean percentage across conditions.

**Gridness in trapezoid environments**. To simulate the effect of asymmetric boundaries in a trapezoid enclosure on gridness[36], we took cluster positions from simulations after learning in square environments, and ran an additional learning phase for 250,000 trials. In this new learning phase, the shape of the environment was now a trapezoid (the agent could only move to those locations), and the annealed learning rate schedule continued (starting at 0.0025, reducing to 0.002 at the end). The trapezoid dimensions were $5 \times 24 \times 50$ pixels, closely matching the proportions in[36] ($0.2 \times 0.9 \times 1.9$ meters; multiplied by (50/1.9) equals to 5.26, 23.7, and 50).

In order to test whether the asymmetric boundaries of the trapezoid affected gridness, the trapezoid was split into two halves and we computed the grid score for the spatial autocorrelogram on the left (wide) and right (narrow) side of the shape. Due to discretization, we split it as close to equal as possible. The wide half extended from the leftmost pixels to the 17th pixel (338 pixels), and the narrow side extended from the 18th pixel to the 50th pixel (339 pixels).

Due to the asymmetrical shape of the trapezoid environment, the procedure for generating a movement trajectory above leads to a slightly biased sampling of the wide part of the trapezoid, and less exploration of the middle and top parts of the shape. To deal with this, we made a slight change to the possible steps after generating a step that brings the agent out of the environment, described below. For each trial, the step was generated as before. If the generated step was out of the environment, the step was cancelled, and the next step was determined as follows. If the step generated would have brought the agent out of the bottom of the trapezoid, the next step was sampled from [0, 0, 1, 1] (stay or up). If the step brings the agent out to the top, the next step was sampled from [−1, −1, 0, 0] (down or stay). When the step takes the agent out of the left of the trapezoid, then the next step to be sampled on the horizontal axis were [0, 1, 1, 2, 4], towards the inner portion of the environment. If the step took the agent out of the right side of the trapezoid, the next step was generated as before, from [−4, −2, −1, −1, 0, 1, 1, 2, 4]. This is because when the agent is out of the trapezoid on the horizontal (left-right) axis, the agent could still be in the middle of the shape on the vertical axis, since the shape becomes more narrow as it reaches the right. Finally, when it lands exactly in the middle of the horizontal axis, but is out of the shape (on the horizontal axis), the next step to be sampled from on the vertical axis is [−1, 0, 1].

**Reporting summary**. Further information on research design is available in the Nature Research Reporting Summary linked to this article.

## Code and data availability

All simulation code and plotting scripts are available at https://github.com/robmok/code_gridCell. Data generated and used for this manuscript is available at https://osf.io/2dz3x/.

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

## Acknowledgements

We thank all members of the LoveLab for valuable input. We thank Roddy Grieves for his advice on analysis of grid measures and members of the Institute of Behavioural Neuroscience (IBN) at UCL for valuable input and discussions. This work was funded by the National Institutes of Health [grant number 1P01HD080679]; a Royal Society Wolfson Fellowship [18302] to Bradley C. Love; and a Wellcome Trust Senior Investigator Award [WT106931MA] to Bradley C. Love.

## Author contributions

R.M.M.: conceptualization, formal analysis, methodology, software, visualization, writing—original draft, writing—review & editing. B.C.L.: conceptualization, formal analysis, funding acquisition, methodology, resources, supervision, visualization, writing—original draft, writing—review & editing.

## Competing interests

The authors declare no competing interests.
