## [Peer Review File · Nature Communications]

Editorial Note:

Comments received from an additional expert during the first round were not used to reach the decision and thus are not reported.

Reviewers' Comments:

Reviewer #2 (Remarks to the Author)

Review Mok

Mok and Love present a novel account of place and grid cell firing based on concept learning models. The proposed key argument is that clustering-based concept learning is a fundamental learning mechanism that gave rise to the seemingly spatial firing patterns found in the medial temporal lobe, namely place and grid cells. The authors show that a simple clustering model can account for the emergence of grid cells if relatively uniform sampling of the spatial environment is assumed, and that room geometry dependent changes in gridness, which have been empirically observed, are predicted by a clustering model. Their view contrasts with a number of published hypotheses that have argued that spatial mechanisms are employed during concept learning.

This is well written study that addresses an important and hotly debated issue. The proposal is original and in some aspects quite clever. It is also interesting and important to see (radically) different perspectives on this important topic. Nevertheless, I have major reservations regarding the assumptions underlying their model and the ability of the theory to explain several findings in spatial navigation research. I therefore remain doubtful it represents a viable model of place and grid cells.

Major points:

- First, the authors overly generalise from a model that shows grid like responses to a model that also shows place cells. While their title and abstract mention place cells, their paper focusses on the emergence of (equidistant) grid like responses if environments are randomly explored. It seems unclear how, given the same random exploration, their model would also predict place cells.

- The model makes the strong prediction that grid cell emergence is dependent on the exploration policy (random/uniform versus constrained). Depending on whether sampling is random or constrained the *same cells* would behave place cell or grid cell like, a phenomenon not observed to the best of my knowledge. I am aware that a variety of spatial information is encoded in (medial) entorhinal cortex, but the idea that these are the same populations of cells does not seem warranted.

- The model also seems to predict that grid cells emerge on the same time scale as place cells, unlike empirical results (Hafting et al. 2008). The model uses simulations with 1 million iterations, while it is known that grid cells emerge within seconds in any new environment.

- The model also seems to make the prediction that throughout learning the locations of the clusters, and hence the firing fields, constantly shift with new experience, a phenomenon not observed empirically to the best of my knowledge. The authors moreover assume distinct learning and “after learning” phases in which positions are not updated any more, but it remains unclear on what empirical or theoretical grounds this procedure would be warranted.

- None of the other major distinction between place and grid cells, such as (lack of) remapping effects, walking direction dependence, anatomical distinctions, effects of silencing of one cell type on the other, developmental differences etc., are accounted for. I don't want to hold the authors to the overly high standard of having to explain everything, but their account seems to assume that place and grid cells are the very same mechanism that only seems to look different under different sampling scenarios. I think this is not a position that is supported by existing studies.

Reviewer #3 (Remarks to the Author)

Summary:

In this manuscript, the authors explain how the k-means algorithm will converge on hexagonally arranged cluster centers in 2D spatial domains and on cluster centers in categorically clustered environments. They use this to argue that the hippocampus has a role in finding conceptual clusters, and hypothesize that grid cells are monitoring error of the proposed solution. The authors show results on

simulated data from random walks in square, circular, and trapezoidal environments. They illustrate how hexagonal griddiness over cluster centers emerges from applying k-means over time in the square + circular environments, and show that griddiness remains higher in the wider end of the trapezoid, capturing an experimental finding. The main insight of this paper is to demonstrate that an algorithm that supports categorical concept learning could generate gridlike representations as well, suggesting that perhaps spatial representations are a special case of a system that learns conceptual spaces more generally. The authors argue that the hippocampus' primary role could therefore be concept learning, and that spatial learning is a special domain in which categories arrange themselves hexagonally.

Major comments:

The ideas put forward in this paper are quite interesting and have the potential to be impactful. However I think some important theoretical and empirical context is still missing, and the connection with data is still too sparse to draw conclusions.

Major comments:

(1) Some references are missing clarifying what was already known about the connection between the clustering interpretation and the geometric underpinnings of k-means (centroidal Voronoi tessellation, which is well known to produce hexagons in 2D, e.g. Du et al (1999)). The connection between grid cells' hexagonal lattice and the optimality of hexagonal lattices for space-filling in self-organized maps has also been explored in literature (e.g. Stella et al 2015), which is what k-means is doing here under the hood. This will help clarify which are the novel contributions of this paper and what is its main message.

(2) It seems like the authors interpret their results to mean that that clustering algorithms can produce spatial results, and so hippocampus is intrinsically for clustering. But k-means is a geometric algorithm that will try to space cluster centers at an even distance from each other, so you could also think of it as metric algorithm in which categorically structured data is the edge case. You could also think of this solution as a constrained way to maximize variance explained (Gaussian mixture model interpretation). In short, I'm not sure how this relates to other ideas about cognitive maps, metric representations, transition statistics, etc. and what role the authors imagine for hippocampus.

(3) The connection to empirical data is thin. A strength of this model is that it could make some pretty specific, testable predictions about how boundaries affect grid cells, and how grid cells change over the course of learning, and how place cells and grid cells should change together. For example: other results measured by Krupic et al (2015), time courses of rescaling in Barry et al (2007), local remapping in Krupic et al (2018), fragmentation in Derdikman et al (2009), long-term remapping in Carpenter et al (2015).

Perhaps the authors are not trying to make strong claims about the details of grid fields, but if not the authors should be clear about why not.

(4) What basis is there for MEC, rather than LEC, supporting concept learning, or having categorical representations? The extent to which MEC has object representations still seems to be position with respect to object (Hoydal et al 2018). Some discussion of how this model fits into the current views on LEC/MEC dichotomy would be useful.

(5) I don't know of evidence that grid cells are used in error updating in the ways the authors hypothesize. It seems like this model could be used to make some specific predictions about how grid cells should change with experience, and how place and grid cells should change and remap together. I think it is important that the error updating part of this hypothesis be connected to existing data or predictions, otherwise it's not obvious under what conditions this theory should be preferred over the interpretation that grid field centers are trying to space themselves evenly over the conceptual space (which would be more of a metric representation interpretation than a concept learning one). Also, is there any evidence grid cells become more griddy over the course of an experiment as they do in the paper's simulations?

Minor comments:

(1) Place cells are generally thought of as representing concepts in an episode-specific way, making them an unlikely substrate for concepts that transcend multiple episodes. How do the authors think about this?

(2) Do the authors think grid cells are parameterized by k in the same way as k -means, or is this a simulation assumption?

(3) Why is clustering useful for continuous, non-categorical concept spaces? Discrete clusters are a natural basis of generalization in category learning but not for continuous spaces in which all points are valid possibilities.

(4) I'm surprised the range of griddiness scores are as large as they are. It seems like hexagonal arrangement should definitely be optimal (at least in the circle, maybe not the square) and that this should be what k -means ultimately converges to in 2D. Do the authors have thoughts on this? Is it sensitive to optimization parameters?

(5) I was also unsure about the fraction of simulated cells that are gridlike reported by the authors (Supplemental Results). On what parameters does that fraction depend and for what parameter values does it hold?

(6) Why would there be multiple grid cell modules at different spatial scales, and multiple phases within module? These are generally thought to help disambiguate location, allowing location to be uniquely decoded from grid cells, but might serve a different purpose in this model if grid cells are for error monitoring, not location indexing.

Other:

This paper (Whittington et al 2018) recently showed that place cells tend to remap in new environments to positions that align with a firing field of the grid cell. They had a different interpretation of that result, but it seems potentially consistent with your model as well.

Responses to reviewers

We thank the reviewers for their positive and constructive comments.

In order to address the reviewers' concerns, we would first like to clarify the purpose of our model and this manuscript, and what insights our study provides. Our model, in terms of levels of analysis, is a high-level model. Our clustering models in the lab were based on ideas on learning and memory, developed to capture behaviour in concept learning tasks, and later also captured neural data in fMRI. Strikingly, we show here that these types of models can even capture cellular firing patterns in spatial experiments (in fact, better than other models; see below), indicating how seemingly different processes could arise from similar computational principles. The central idea is that 'place' and 'grid' cells are not intrinsically nor specifically spatial, and that space is not the basis for encoding conceptual knowledge (opposed to other views; e.g. Buzsaki & Moser, 2013, *Nature Neuroscience*; Bellmund et al., 2018, *Science*). Rather, we propose that there is a general learning system that is the basis of both spatial and conceptual tasks. This is a novel position we put forward with our model, which goes beyond previous work that merely describes or points out a superficial link between the spatial, conceptual, and memory literatures (e.g. all involve the hippocampus). We believe these ideas will have major implications for studying and interpreting experimental findings in the future. We have clarified these points in manuscript.

Our perspective comes from a higher level to provide a general theoretical framework covering disparate experiments and tasks. All models in some way abstract away from the details, and our model is high level in comparison to other place and grid cell models since it attempts to capture general principles, rather than specific phenomena in spatial environments. For example, clusters are abstract and are used in the model to build an internal representation of the task or environment - they are not intended as a one-to-one mapping to individual place (or concept) cells. The model represents the abstracted, general mechanisms that could be taking place (see responses below for further details). Almost all other place and grid cell models are low-level spatial models, which account for data in spatial experiments, but have less explanatory power when generalizing to other tasks or when abstract and higher-dimensional spaces are considered. Our model makes predictions for spatial cases and extends to abstract spaces, and also accounts for many issues the reviewers raised with respect to known empirical findings, which we clarify below and in the manuscript with an extended discussion.

Please consider our comments below in light of what our model and account is trying to achieve.

Reviewer 2

1. First, the authors overly generalise from a model that shows grid like responses to a model that also shows place cells. While their title and abstract mention place cells, their paper focusses on the emergence of (equidistant) grid like responses if environments are randomly explored. It seems unclear how, given the same random exploration, their model would also predict place cells.

Thank you for this comment. We clarify our model and how place cells fit into our account. The high-level abstract aspect of our model should be noted.

Our account includes both place and grid cells, and in fact place cells are more central than the grid cells for representing concepts. We think of place cells as the spatial analogue of concept cells (Quian Quiroga et al., 2005, *Nature*; Quian Quiroga, 2012, *Nature Neuroscience*), so a place cell codes for a particular location in a given spatial environment, like a concept cell codes for a particular concept in representational space (the abstract or feature space that concepts lie in). Grid cells emerge from monitoring place cells. Specifically, our model says that, a system (or cell) monitoring a set of place cells (e.g. in a square environment) will self organize and produce solutions with grid like patterns. We call this a "cluster match" or "monitoring error", which we realise was not clearly explained. To clarify this point, we explain what error monitoring means in the context of clustering models.

When an agent visits a new location, the closest cluster will activate. An individual cluster (or place cell(s)) would know if the location is close to itself and can compute its distance from that location (error from cluster). However, if it was not the closest cluster, it does not activate, and so it has no information. A (grid) 'cell' monitoring many of the clusters (place cells) can tell how close or far away the current location was from *any* cluster - and therefore can monitor error *in general*. This is exactly the (inverse of the) entropy term used in clustering models (Love et al., 2004, *Psych Review*; Anderson, 1991, *Psych Review*), and can be informative of whether there is a cluster or concept match among all existing clusters. Therefore, grid cell and non-grid spatial cell activations of multiple fields are precisely what we suggest represents the (inverse) error, or the degree of match to any existing cluster.

Note that a cluster should not be thought of as a single place cell (not a one-to-one mapping). Rather, a cluster should be thought of as a group of similarly tuned cells that can be described by the cluster at a higher level of analysis. There may be many place cells that correspond to a cluster (e.g. there is more than one cell that responds to individual concepts - that is, 'concept' cells rather than a single 'grandmother cell'; Quian Quiroga, 2012).

The cluster locations and the distance between clusters (e.g. between and within categories in a concept learning task) builds up the representational / internal space of the model, which can be linked to the brain (e.g. Mack et al., 2016; also see Mack et al., 2013; Davis et al., 2012 from our lab).

Place cells are therefore a key part of our account. However, we take the reviewer's point that it is unclear how the model predicts place cells as meaning it is unclear how they emerge, which we clarify here. The clustering algorithm is based on competitive learning accounts (commonly used in learning problems), and is a Kohonen learning rule that has been used to explain how cells can selectively respond and come to form the kind of response properties such as those found in place cells. Our model simply states that cells start out with some tuning (random initialization and get recruited or move their tuning in response to inputs in a competitive way (winner-takes-all in our case).

We realise this was not sufficiently explained in the manuscript and added clarification in the introduction (p4-5):

“For example, in a simple case with two clearly separable and internally coherent sets of objects, a clustering model would use one cluster to represent each concept, each of which would be centered amidst its members in representational space (e.g. Fig 1A). When the model is presented with a novel item, the closest cluster in representational space is activated, which signals the category membership of the item. An error-monitoring signal gauges how well an item matches this closest cluster in representational space. Because these models are winner-take-all (i.e., only one cluster maintains a non-zero output) and the item is assigned to this most likely cluster during the learning update, the In these models, only the closest cluster maintains non-zero activation (winner-takes-all), so an error-monitoring signal (entropy term) ‘monitors’ activation of all existing clusters which can tell us how close or far away the current location was from any cluster, which acts as a cluster match (or non-match signal) is driven by the activity level of the winning cluster^{16,17}. These clustering representations successfully capture patterns of activity in the MTL^{18,19} and are in accord with the notion that the human hippocampus contains “concept” cells in which individual cells respond to a specific concept, much like how a cluster in a possibly high-dimensional space can encode a concept²⁰. Analogously, place cells respond to a location in a particular two-dimensional spatial context. It is important to note that clusters are abstract entities in the model, and there need not be a one-to-one mapping to single concept or place cell (e.g. a cluster can correspond to a group of place cells with similar tuning).”

2. The model makes the strong prediction that grid cell emergence is dependent on the exploration policy (random/uniform versus constrained). Depending on whether sampling is random or constrained the *same cells* would behave place cell or grid cell like, a phenomenon not observed to the best of my knowledge. I am aware that a variety of spatial information is encoded in (medial) entorhinal cortex, but the idea that these are the same populations of cells does not seem warranted.

First, to clarify, we propose that grid cells (in the medial entorhinal cortex) monitor place cells (in the hippocampus; see response to comment 3 below for empirical evidence). We focus on these brain regions strongly implicated in spatial processing and memory, and propose that they may rely on the same general-purpose algorithm. We hope the clarification of the model in above in response 1 helps.

Second, we propose that grid-like firing patterns are affected by the properties of the environment (not exploration ‘policy’, see next point) - that when the environment is a symmetric, uniformly traversable environment (e.g. square or circle), spatial cells in the mEC will look grid-like. In other cases, such as in a maze, ‘grid cells’ - do not show hexagonal firing fields (e.g. Derdikman et al., 2009, *Nature Neuroscience*; hairpin maze). When the environment is manipulated, the same ‘grid’ cells that are grid-like in a square and non-grid like in the trapezoid enclosure (Krupic et al., 2015; *Nature*; Krupic et al., 2018; *Science*).

Please note that ‘exploration’ in the model should not be interpreted in the sense used in reinforcement learning, but simply input of the data fed to the learning system. The experiment design determines what the input is in category learning, and it indirectly does the same in the spatial foraging task (with no specific goal). None of the results depend on the exact assumptions of exploration, only that these results will fall out of unconstrained spatial tasks with simple, symmetric environments (e.g. square or circle).

3. The model also seems to predict that grid cells emerge on the same time scale as place cells, unlike empirical results (Hafting et al.2008). The model uses simulations with 1 million iterations, while it is known that grid cells emerge within seconds in any new environment.

First, we would like to clarify that in our account, there are no intrinsic ‘place’ or ‘grid cells’, rather there is a system in which the population of neurons code for the task at hand, and spatially-selective cells emerge from the spatial properties of the experimental setup. We specify how the receptive properties of these cells should develop depending on the task or environment, and make a number of correct predictions. Since we are putting forward a single account for spatial and non-spatial learning system, one would not expect us to get everything right. However, our account does match up to empirical findings for the timing of grid cell development, which we describe below.

In our model, grid cells monitor (connected to and receive input from) place cells, and then self organize over time to produce a hexagonal firing pattern. This is consistent with developmental work (Wills et al., 2010, *Science*; 2012, *Frontiers in Neural Circuits*), where place cells (and other spatial cells) appear in baby rats very early in life, then grid cells develop shortly after. Note the high-level commonalities between the empirical data and the model, that the grid cells don't have grid-like firing fields immediately but only with input from place cells, and that the model captures the temporal order. In adult rats, grid cells emerge much quicker in new environments, likely related to prior knowledge of spatial environments.

Relatedly, our model captures the finding that place cells are causally involved for the activity of grid cells. This is strongly supported by evidence that inactivation of the hippocampus (with place cells) leads to grid cells in *mEC losing the periodicity of their firing fields* (Bonnievie et al., 2013, *Nature Neuroscience*), whereas inactivation of the mEC (with grid cells) only mildly affect place cell firing fields in the hippocampus (Hales et al., 2014, *Cell Reports*).

Finally, with respect to time in modelling in general - the number of trials in simulations don't match up exactly with 'real-time', so the numbers should not be taken literally. We run more 'trials' (in this case, steps) to ensure a stable solution, as is commonly done with simulations (and it turns out it doesn't matter much). With respect to the reviewer's concerns: note that the number of updates (batch update) is significantly lower than the number of trials, and the grid-like representations actually emerge relatively quickly (see figure S1).

We have added clarifications and discussion on the temporal and causal aspects of the model to the manuscript (p15):

“Our model also captured the causal relation between place and grid cells. In our account, grid cells play a cluster-match or error-monitoring function where they monitor (connected to and receive input from) place cells, and self organize over time to produce a hexagonal firing pattern (also see ³⁶). This is consistent with developmental work^{40,41}, where place cells appear in baby rats very early in life, and grid cells develop shortly after. The model also predicts that place cells are causally involved for the activity of grid cells. This is strongly supported by empirical work showing that inactivation of the hippocampus (with place cells) leads to grid cells in

mEC losing the periodicity of their firing fields⁴², whereas inactivation of the mEC (with grid cells) only mildly affect hippocampal place fields⁴³

4. The model also seems to make the prediction that throughout learning the locations of the clusters, and hence the firing fields, constantly shift with new experience, a phenomenon not observed empirically to the best of my knowledge. The authors moreover assume distinct learning and “after learning” phases in which positions are not updated any more, but it remains unclear on what empirical or theoretical grounds this procedure would be warranted.

This is a good question and we thank the reviewer for the opportunity to clarify our position. We stress the higher-level aspect of the model when considering the link between clusters with place and grid cells. A cluster, like latent variables in cognitive models, is abstract, and so a cluster can correspond to a group of place cells with similar tuning. To answer the reviewer’s comment, we first clarify how the clustering model learns and how this links to place and grid cell firing fields. Then, we refer to the supporting empirical evidence that firing fields do change over time and with learning.

In concept models, the learning process involves clusters 'tuning' their receptive fields, in which they find where in representational space they should lie and hence tuning their response to certain feature dimensions. In the model presented here, cluster tuning (shifting) reduces over time with learning (learning rate reduces). The model updates more at the start when it needs to learn (error is high) and updates less when it has learnt or ‘understood’ the environment (error is low). It is a standard property for learning models to update more when there is more error and less when there is less error, i.e. it is task driven. Our account suggests that it is a general learning system that underlies these effects rather than specifically spatial.

We suggest that place and grid cells do this early in learning, with smaller changes or shifts later. As the agent (or rat) learns or grasps the spatial environment, the internal representation stabilizes - which manifests in our model as a reduction in the learning rate, and stabilizing of the clusters as they settle (as this is what the algorithm does). Therefore, our model predicts firing fields might shift early but stabilize over learning. There is emerging evidence that place and grid cells do show changes over time with learning (Sun & Giocomo, 2018, SfN abstract 604.25, who showed that grid cell fields shift early in learning and stabilize over time (“spatial learning”) in the same environment, and Ziv et al., 2013, *Nature Neuroscience*, for evidence that place cells do show some change of spatial properties over days in the same environment).

Furthermore, there is empirical evidence that even later in learning (e.g. grid cells found in a square) firing field of grid cells to shift in trapezoids (Krupic et al., 2015, *Nature*) and in gradually changing environments (Krupic et al., 2018, *Science*). There is also emerging work that the grid fields closest to rewards actually shifts toward rewards (Boccarda & Csicsvari, 2018, Grid Cell and Cognitive Maps Meeting at UCL abstract), predicted by our model.

“The authors moreover assume distinct learning and “after learning” phases”

We used the terms "before" and "after" learning to mean early and later in learning for convenience (plotting and illustrating the what happens with learning), where later in learning the cluster positions have stabilized and shift less (lower learning rate).

5. None of the other major distinction between place and grid cells, such as (lack of) remapping effects, walking direction dependence, anatomical distinctions, effects of silencing of one cell type on the other, developmental differences etc., are accounted for. I don't want to hold the authors to the overly high standard of having to explain everything, but their account seems to assume that place and grid cells are the very same mechanism that only seems to look different under different sampling scenarios. I think this is not a position that is supported by existing studies.

The goal of our paper was to provide an account to link spatial and concept / memory findings in a way that goes beyond mere description (as others in the past have done), which we achieved. Although we don't claim to have explained everything, we think our model captures a key mechanism of the hippocampal-entorhinal circuit during learning by showing that the way in which spatial information is learnt in the brain (place/grid cells) shows a striking parallel with the organization of abstract information in concept learning. This high level consideration is key in considering what this model provides. In addition, we do capture empirical data as detailed in the responses above.

We offer an account of how spatial and conceptual representations may work under similar mechanisms, particularly early in learning, and lay the foundation for future work to tackle questions that link spatial, conceptual, and memory processes. For example, there could be consolidation processes such as those proposed in the memory literature, where neural representations change over a different timescale. We think that there is a coupling between place and grid cells as firing fields and properties are established, but we are agnostic on how that coupling changes over extended time, such as after sleep. Future work should consider these interesting questions.

Furthermore, we acknowledge that there are other mechanisms important in representing space - head-direction cells, unidirectional place cells, vestibular signals, and other signals certainly play a role in representing the environment, and could work under different mechanisms and principles. It could be that some place cells play more 'abstract' roles such as learning or are sensitive to context (e.g. those that remap versus those that do not) whereas others may be more sensitive to specific spatial properties (e.g. distance from a border). Future work should consider these questions such as remapping and heading direction and how this links with concept processing. We think that our work raises many interesting questions and hope that others will join us in tackling these questions from multiple perspectives.

In the manuscript, we extended the discussion to clarify our perspective. We also suggest future directions (in addition to the existing final paragraph):

"We offer an account of how spatial and conceptual representations may work under similar mechanisms, particularly early in learning, and lay the foundation for future work to tackle questions that link spatial, conceptual, and memory processes. For example, there could be consolidation processes such as those proposed in the

memory literature, where neural representations change over a different timescale. It would also be interesting to examine how findings in the spatial domain such as remapping effects and head-direction signals relate to more abstract domains, such as by examining mixed-selectivity and task representation during complex cognitive tasks across species. We anticipate much exciting work across fields tackling how the brain organizes information for knowledge and adaptive behavior.”

We thank the reviewer for the thoughtful comments.

Reviewer 3

Major comments:

The ideas put forward in this paper are quite interesting and have the potential to be impactful. However I think some important theoretical and empirical context is still missing, and the connection with data is still too sparse to draw conclusions.

We thank the reviewer for the positive comments, and we have attempted to address all concerns below.

Major comments:

(1) Some references are missing clarifying what was already known about the connection between the clustering interpretation and the geometric underpinnings of k-means (centroidal Voronoi tessellation, which is well known to produce hexagons in 2D, e.g. Du et al (1999)). The connection between grid cells' hexagonal lattice and the optimality of hexagonal lattices for space-filling in self-organized maps has also been explored in literature (e.g. Stella et al 2015), which is what k-means is doing here under the hood. This will help clarify which are the novel contributions of this paper and what is its main message.

Thank you for pointing this out. We have added some references and clarified the novel contributions of our account.

Indeed, circle packing in two-dimensions is known lead to hexagonal solutions, and grid cell researchers often point to the optimality of the hexagonal arrangements in grid cells to fill spatial environments. We provide clarification on the key contributions of our account below.

One view suggests that the brain's spatial machinery (such as grid cells) may be relied upon for coding abstract, conceptual spaces (see Bellmund et al., 2018, *Science* for a recent theoretical proposal), where the hexagonal code acts as a 'universal metric for space' (e.g. McNaughton et al., 2006, *Nature Neuroscience*) and can generalize to other spaces. However, there are many cases where the hexagonal code distorts, such as in trapezoids (Krupic et al., 2015, *Nature*). Another question is what could the metric be for a high-dimensional feature space for more abstract or conceptual tasks? Why would a hexagonal code make sense, especially when not all the space is equally relevant (in concepts or task spaces)?

In contrast, we suggest that the MTL system is a general learning system that builds a representation of the current task or environment. The hexagonal code falls out of a learning algorithm in specific 2D cases. When it is a 2D space such as a physical

square or circular enclosure, this brain system shows cells that are tuned to spatial properties such as place and grid cells. When the environment (or task context) is one where relevant parts of the space (physical or conceptual) is not uniformly explored or explorable (a maze or a clumpy conceptual space; e.g. see our Fig 1A), these brain areas code for the relevant aspects of the environment (or task context), but the representation is no longer a hexagonal code (also see Davis et al. 2014, *Journal of Neuroscience*).

We think that the empirical work is more in line with our account where the hippocampal-entorhinal circuit flexibly learns the environment using an algorithm such as clustering, rather than force a hexagonal code onto the structure of the environment (see responses below and to reviewer 2 for evidence).

We contribute an account of a general learning mechanism that captures conceptual learning and demonstrates how ‘spatial’ cells such as grid cells are a specific case of learning and representation of a 2D environment, in contrast to some of the other ideas in the literature (e.g. Bellmund et al., 2018). As mentioned at the start of our response letter, this is the novel account we put forward with our model, which goes beyond previous work that merely describes or point out a superficial link between the spatial, conceptual, and memory literatures.

We have added additional citations and clarification in the introduction and discussion (p14):

“Our proposal stands in contrast to other ideas that a dedicated, phylogenetically older spatial navigation system in the MTL supports the newer, higher-level cognitive functions^{5,11}. In particular, we suggest there are no intrinsic ‘place’ or ‘grid’ cells, but instead a flexible system that will represent the relevant variables at hand, including physical space. Emerging evidence shows that cells in the MTL exhibit mixed-selectivity in that they respond to multiple variables, such as place and grid cells that also code for task-relevant sound frequency¹⁴, routes^{31,32}, objects and context³³, and time³⁴, suggesting a flexible code.”

(2) It seems like the authors interpret their results to mean that that clustering algorithms can produce spatial results, and so hippocampus is intrinsically for clustering. But k-means is a geometric algorithm that will try to space cluster centers at an even distance from each other, so you could also think of it as metric algorithm in which categorically structured data is the edge case. You could also think of this solution as a constrained way to maximize variance explained (Gaussian mixture model interpretation). In short, I’m not sure how this relates to other ideas about cognitive maps, metric representations, transition statistics, etc. and what role the authors imagine for hippocampus.

First, we note that we are not doing k-means, but there is a close connection. This is an online learning algorithm based on work in the memory literature and hippocampal function, which is used to model spatial and concept learning (Love et al., 2004).

Clustering is a form of learning. The hippocampus is involved in learning, and our work on concept learning suggests that this brain region plays a role in clustering concepts (e.g. Mack et al., 2016, *PNAS*). We show that clustering can produce spatial representations found in the brain, which means the MTL circuit can use this

approach to generate grid cells. Clustering is an effective method for learning and producing useful representations, so we think it is a plausible way the hippocampus does spatial and concept learning.

We are not claiming that the hippocampus **only** performs clustering – there are other functions of the hippocampal formation that we do not touch on. For instance, there are head-direction cells, vestibular signals, and other cells that play other roles, which are separate to the mechanisms we are focusing on.

Added to the discussion (p14):

“Clustering is a flexible mechanism and can learn many representations in multi-dimensional space. It is an appropriate mechanism for organizing multi-modal, complex information for consolidation of knowledge for memories and concepts. Much evidence suggests this system as a general learning system for organizing knowledge into a useful representation, which can be used for effective behavior and for memory consolidation.”

"But k-means is a geometric algorithm that will try to space cluster centers at an even distance from each other, so you could also think of it as metric algorithm in which categorically structured data is the edge case."

We are modelling learning and representation (of the environment or concept space). We know that humans can learn concepts, the hippocampus is involved in learning and has concept cells, and clustering for categories (in a potentially high-dimensional space) makes sense; it not the edge case. On the other hand, spacing the clusters makes sense for uniformly tiled space, which leads to hexagonal patterns for 2D spatial cases. It is also a degenerate solution from the clustering perspective, as it reveals no hidden structure in the domain. Our model captures both, and what we propose is that it is a flexible system that can do both. The higher level abstract contribution is key here.

“You could also think of this solution as a constrained way to maximize variance explained (Gaussian mixture model interpretation).”

Our account is consistent with a Gaussian mixture model interpretation, since clustering on 2D space is a special case of GMM (with equal variances for each Gaussian). Considering our high-level goal, this doesn't change our account's interpretation.

"In short, I'm not sure how this relates to other ideas about cognitive maps, metric representations, transition statistics, etc. and what role the authors imagine for hippocampus. "

Thank you for this opportunity to allow us to explain and explore this.

Our lab has developed and used clustering models of concept learning, which have been applied to conceptual spaces. These models have successfully captured concept learning behavior and neural representations (Love et al., 2004, *Psych Review*; Sakamoto & Love, 2008; Davis et al., 2012, *JEP:LMC*; 2012; *Learning and Memory*; 2014, *Journal of Neuroscience*; Mack et al., 2016, *PNAS*, 2017; *PRNI*; 2017, *bioRxiv*, Braunlich & Love, 2018, *bioRxiv*). We are bringing these ideas over to

space, suggesting similar underlying mechanisms. These papers on clustering models have been closely related to these other ideas on the cognitive map. Our current study would connect with these ideas, introducing a new literature to recent ideas on the cognitive map.

We briefly explain how clustering models represent concepts and how they relate to concept cells in the hippocampus and other ideas on the cognitive map. In clustering models of concept learning (e.g. Love et al., 2004, *Psych Review*), the model assigns cluster membership (and so category membership) to a stimulus by its similarity to a cluster in feature space (closest cluster in multidimensional feature space). When the model is presented with a novel item, it can assess which is the closest cluster and assign category membership. This is closely related to concept cells in the hippocampus, where individual cells respond to a specific concept, like a cluster inhabiting a high-dimensional space (a conjunction of many features that represent that concept). This is also consistent with many other ideas of hippocampal function, such as relational memory and temporal associations (Eichenbaum) and statistical learning (Schapiro, Turk-Browne) and transition statistics (Successor Representation; Dayan, 1993), with all relevant objects or memories arranged in a map form.

We added more details to our introduction of clustering models in the introduction (p4), and added a discussion of our account with other ideas on the cognitive map (p16):

“Clustering models organize information and represent concepts in feature space in a way that enables identification, classification, and generalization to novel objects¹⁶. These models are closely linked with ideas on episodic memory²¹ and accounts of hippocampal function including relational memory⁴⁴, statistical learning⁴⁵, and transition statistics (successor representation^{35,46,47}), with objects or memories arranged in the form of a cognitive map. Our view is that the formation of the map is borne out of a flexible, general learning system supported by the MTL. We offer an account of how spatial and conceptual representations may work under similar mechanisms, particularly early in learning, and lay the foundation for future work to tackle questions that link spatial, conceptual, and memory processes.”

(3) The connection to empirical data is thin. A strength of this model is that it could make some pretty specific, testable predictions about how boundaries affect grid cells, and how grid cells change over the course of learning, and how place cells and grid cells should change together. For example: other results measured by Krupic et al (2015), time courses of rescaling in Barry et al (2007), local remapping in Krupic et al (2018), fragmentation in Derdikman et al (2009), long-term remapping in Carpenter et al (2015). Perhaps the authors are not trying to make strong claims about the details of grid fields, but if not the authors should be clear about why not.

First, the key purpose of our paper is to provide a new account that attempts to capture a learning mechanism that can explain aspects of neural representations of physical space and concept space. Our major contribution is a novel theoretical proposal that unites a large number of findings across domains through a well-specified mechanism, and is not based on spatial mechanisms (as others have proposed).

Second, our model actually does a good job in capturing grid cell properties compared to existing models. For example, other models either do not capture the hexagonal code (Stachenfeld et al., 2017, *Nature Neuroscience*; 90 degree grids) or need to build in additional constraints (Dordek et al., 2016, *eLife*; non-negativity constraint, which changes the 90 degree grids to 60 degree grids). When they show a large proportion of grid cells, they are *too* good in that all the simulated cells are grid cells (e.g. Dordek et al., 2016; Krupic et al., 2014, *Philosophical Transactions of the Royal Society B*). In our model, we show, without much tuning, a good match of the proportion of grid cells with empirical data.

We also capture the temporal and causal link between of place and grid cells, where place cells appear first then grid cells in development (see response 3 to reviewer 2 above for more details). With this, a low learning rate after learning means that big changes don't occur, although minor changes still occur (e.g. trapezoid manipulation). Future modelling work will address the issues of changing environments, and how place and grid cells change over time.

As the reviewer points out, our main goal is not to match the details of place and grid cells, but show that our model can capture certain crucial aspects of spatial and conceptual learning, and provide a general conceptual framework to guide future work.

We have now added these points to the discussion:

“Our high-level account provides a general theoretical framework applicable to a range of tasks, in contrast to lower-level models of place and grid cells which give predictions in specific spatial contexts but have less explanatory power to generalize across contexts. The model’s contribution is providing the abstracted, general mechanism that could be used across domains. Despite the high-level aspect of our model, we were able to closely match a number of empirical observations from the spatial literature. Our model showed a similar proportion of grid-like cells in found in mEC. Other models either do not capture the hexagonal code (³⁵; 90 degree grids) or need to build in additional constraints (³⁶; non-negativity constraint changes the 90 degree grid patterns to 60 degree grid patterns). When they find a large proportion of grid cells, they are too good in that all the simulated cells are grid cells³⁶ (also see ³⁷). Here, we used a simple model, from a high-level perspective based on ideas from concept learning and memory, and matched the proportion of grid cells with empirical data. Furthermore, our account also suggests that it is unlikely that the MTL codes for abstract spaces using a metric like those found in grid cells. When the representational space is high dimensional and not equally relevant, such as in conceptual or task spaces, distances between objects would not be represented equally. For instance, how a trained ornithologist represents different species of birds (clearly spaced apart brain representations) will be very different to how she represents different species of donkey (relatively close together). In the same way, the internal representation of navigating through a known route will have a different metric to an unfamiliar one³⁸. In support of these ideas, empirical work has shown that grid cells also lose their grid-like properties in more complex environments such as mazes³⁹. Our model also captured the causal relation between place and grid cells. In our account, grid cells play a cluster-match or error-monitoring function where they monitor (connected to and receive input from) place cells, and self organize over time to produce a hexagonal firing pattern (also see ³⁶). This is consistent with developmental work^{40,41}, where place cells appear in baby rats very

early in life, and grid cells develop shortly after. The model also predicts that place cells are causally involved for the activity of grid cells. This is strongly supported by empirical work showing that inactivation of the hippocampus (with place cells) leads to grid cells in mEC losing the periodicity of their firing fields⁴², whereas inactivation of the mEC (with grid cells) only mildly affect hippocampal place fields⁴³.”

(4) What basis is there for MEC, rather than LEC, supporting concept learning, or having categorical representations? The extent to which MEC has object representations still seems to be position with respect to object (Hoydal et al 2018). Some discussion of how this model fits into the current views on LEC/MEC dichotomy would be useful.

We do not think that mEC plays a role in concept learning and LEC plays no role. The reason we focus on mEC is because our model found that grid-cell like representations could arise from a concept learning-based clustering account, and our point is that spatial representations such as those in mEC are not necessarily specific to spatial representations. Furthermore, we do find evidence of an error-monitoring (entropy) signal in the mEC in Davis, Love, & Preston, 2012, *JEP:LMC*). We think our model has captured something about the MTL circuit which could be a general representational principle (which may also be present in other brain regions). Most studies focus on spatial aspects and are restricted in their interpretation to spatial coding.

But this is a good point, since many people are fixated on some brain area they work on. We are not tied to MEC. We think concept learning (and other types of learning, including spatial learning) is distributed across brain areas, including the mEC and hippocampus - in fact, the hippocampus is where the crucial information (clusters) is stored - it's a circuit.

With regard to how our work relates to current views of MEC/LEC, a standard distinction made in the literature is the MEC deal more with spatial information and LEC with object information (Deshmukh & Knierim, 2011, *Frontiers in Behavioural Neuroscience*), however, emerging work suggests it is more complex than this. For instance, there is a lot of object/category information the MEC (work by Eichenbaum and colleagues, e.g. Keene et al., 2016, *Journal of Neuroscience*). Hoydal et al.'s (2018, *bioRxiv*) work that the reviewer raised suggests there is a type of object-spatial coding, but those cells are more specific to the spatial context, where they care about relative object location but not to object properties. From our perspective it is another type of spatial coding, but not directly related to higher level abstract properties.

We are more agnostic about how our model fits with the MEC/LEC distinction. We think that the general learning mechanism works between HPC and entorhinal cortex - but object properties and coding must come from somewhere, such as LEC and perirhinal cortex (both of which play a role in object coding).

We clarify the general high-level principles and what we claim in the discussion.

(5) I don't know of evidence that grid cells are used in error updating in the ways the authors hypothesize. It seems like this model could be used to make some specific predictions about how grid cells should change with experience, and how place and grid cells should change and remap together. I think it is

important that the error updating part of this hypothesis be connected to existing data or predictions, otherwise it's not obvious under what conditions this theory should be preferred over the interpretation that grid field centers are trying to space themselves evenly over the conceptual space (which would be more of a metric representation interpretation than a concept learning one).

Also, is there any evidence grid cells become more gridly over the course of an experiment as they do in the paper's simulations?

To clarify, grid field centres are spacing themselves evenly over space only when the full 2D space is relevant, like in open field experiments. When it's not, then they won't be as perfectly grid like - as shown in empirical data (trapezoid, mazes). Empirical data shows that 'grid' cells in different environments become less grid-like, which is not consistent with a 'universal metric' interpretation.

Our idea of grid cells monitoring error is also consistent with current findings, but we realise that we do not explain the concept of error in clustering model as well as we should have. We include response 1 to reviewer 2 here that explains our model in more detail, and how 'monitoring error' and 'cluster match' entirely fits with the empirical data with regards to grid cells:

“Our account includes both place and grid cells, and in fact place cells are more central than the grid cells for representing concepts. We think of place cells as the spatial analogue of concept cells (Quiari Quiroga et al., 2005, Nature; Quiari Quiroga, 2012, Nature Neuroscience Reviews), so a place cell codes for a particular location in a given spatial environment, like a concept cell codes for a particular concept in representational space (the abstract or feature space that concepts lie in). Grid cells emerge from monitoring place cells. Specifically, our model says that, a system (or cell) monitoring a set of place cells (e.g. in a square environment) will self organize and produce solutions with grid like patterns. We call this a "cluster match" or "monitoring error", which we realise was not clearly explained. To clarify this point, we explain what error monitoring means in the context of clustering models.

*When an agent visits a new location, the closest cluster will activate. An individual cluster (or place cell(s)) would know if the location is close to itself and can compute its distance from that location (error from cluster). However, if it was not the closest cluster, it does not activate, and so it has no information. A (grid) 'cell' monitoring many of the clusters (place cells) can tell how close or far away the current location was from *any* cluster - and therefore can monitor error *in general*. This is exactly the (inverse of the) entropy term used in clustering models (Love et al., 2004, Psych Review; Anderson, 1991, Psych Review), and can be informative of whether there is a cluster or concept match among all existing clusters. Therefore, grid cell and non-grid spatial cell activations of multiple fields are precisely what we suggest represents the (inverse) error, or the degree of match to any existing cluster.”*

Furthermore, we do find evidence of an error-monitoring (entropy) signal in the mEC in Davis, Love, & Preston, 2012, JEP:LMC)

Also, is there any evidence grid cells become more gridly over the course of an experiment as they do in the paper's simulations?

The focus of our model is not on development, but on how the grid-like pattern could form over a learning episode. We do know that grid cells become more griddy over time in the sense that spatial cells in mEC cells develop over time (before place cells) and become more stable and gridlike in baby rats, and (Wills et al., 2010; 2012). They also become more stable over time with spatial learning in adult rats (e.g. Sun & Giacomo, 2018, SfN abstract 604.25).

We note that these are fair questions but are also very fine-grained aspects of a broad new theory we are offering. We propose an account that provides a broad general theory, which captures findings across two fields and raises interesting questions for the field to pursue. Our proposal is a new idea that should be published and considered across fields, which can be assessed over time with new data and perspectives.

Minor comments:

(1) Place cells are generally thought of as representing concepts in an episode-specific way, making them an unlikely substrate for concepts that transcend multiple episodes. How do the authors think about this?

We prefer to think of place cells as concept cells found in the human hippocampus (Quiari Quiroga et al., 2005, *Nature*). However, we also think about how concepts are learnt over time, or one episode at a time (see our review: Mack, Love, & Preston 2017). Building concepts one episode at a time: the hippocampus and concept formation. *Neuroscience Letters*). There is a close relationship between episodic memory and concept formation and these are key questions that our group and other groups are actively pursuing.

(2) Do the authors think grid cells are parameterized by k in the same way as k -means, or is this a simulation assumption?

This is a simulation simplification, and the specific values used are not important to our general point.

(3) Why is clustering useful for continuous, non-categorical concept spaces? Discrete clusters are a natural basis of generalization in category learning but not for continuous spaces in which all points are valid possibilities.

Concepts are not always strictly categorical or continuous - there are concepts which are clearly categorical (e.g. binary features that perfectly predict category membership), and others which vary on the smooth continuum. Clustering also works with probabilistic categories as well (e.g. Anderson, 1991, *Psych Review*). We are not making the case that it should be one or the other; but that clustering is flexible can deal with many different cases.

(4) I'm surprised the range of griddiness scores are as large as they are. It seems like hexagonal arrangement should definitely be optimal (at least in the circle, maybe not the square) and that this should be what k -means ultimately converges to in 2D. Do the authors have thoughts on this? Is it sensitive to optimization parameters?

As noted above, this is not k -means but is closely related to it. The reason for the large range of scores is partly because any clustering algorithm (including k -means)

are susceptible to local minima – there is variability associated with its solution given different starting conditions. Likewise, our learning model will have order effects. Furthermore, it is not always the case that the best arrangement is hexagonal (e.g. see https://en.wikipedia.org/wiki/Circle_packing_in_a_circle). Finally, boundaries make perfect packing impossible. All these factors affect the ‘grid score’ which is also an imperfect measure.

However, it is particularly interesting that the distribution of grid scores is similarly large in empirical data, and we did not plan to match this at all: e.g. see Wills et al., 2010, *Science*, Figure S7B; Bonnevie et al., 2013, *Nature Neuroscience*, Figure S5; see also Diehl et al., 2017, *Neuron*. Again, we match the empirical data better than many existing models.

(5) I was also unsure about the fraction of simulated cells that are gridlike reported by the authors (Supplemental Results). On what parameters does that fraction depend and for what parameter values does it hold?

The fraction of simulated cells that are grid-like does not change too much with changing of parameters. As can be seen in figure 3D and 3E, as the number of clusters increases, the range of grid scores becomes quite similar. With a slower or faster learning rate, or an increase or reduction of the batch size, the grid scores and fraction of ‘grid’ patterns are near identical, results provided in the code and simulated data.

We have noted this in the supplementary results:

“The percentage of grid-like cells show very little difference when the parameters are altered, such as a slower or faster learning rate, or an increase or reduction of the batch size. These results are provided in the code and simulated data.”

(6) Why would there be multiple grid cell modules at different spatial scales, and multiple phases within module? These are generally thought to help disambiguate location, allowing location to be uniquely decoded from grid cells, but might serve a different purpose in this model if grid cells are for error monitoring, not location indexing.

There could be various reasons for this. One important point is that the selection for grid cells is biased - there are many spatial cells with multiple firing field that don't pass the arbitrary threshold of being a 'grid cell'. Importantly, these non-grid spatial cells or irregular spatially selective cells in mEC have as much spatial information as grid cells (Diehl et al., 2017, *Neuron*). Consistent with this, our account suggests that grid cells are not special in representing spatial or concept spaces, consistent with mixed-selectivity in these cells. Our view is that the brain doesn't take only grid cells and compute location - they take populations of spatial cells in order to do a read out.

In addition, it is not clear whether the grid cell properties found in rodents can be matched directly to humans. For one, inputs to the hippocampal formation are much more sensory driven in rodents whereas inputs are from association cortices in primates and humans. It is unclear whether these properties are the same in primates, and early work from the Buffalo lab suggests it might not be (personal communication at SFN 2018).

We have clarified our position in the discussion (p13-14):

“In particular, we suggest there are no intrinsic ‘place’ or ‘grid’ cells, but rather a flexible system that will represent the relevant variables at hand, including physical space. Emerging evidence shows that cells in the MTL exhibit mixed-selectivity in that they respond to multiple variables, such as place and grid cells that also code for task-relevant sound frequency¹⁴, routes^{31,32}, objects and context³³, and time³⁴, suggesting a flexible code.”

These great questions support why this work should be published, so that it can promote further study on open questions (such as more focus on non-grid spatial cells, their relation with more abstract task or environmental properties). Our proposal is a simple one that will be useful for experimentalists to critique and evaluate against.

Other:

This paper (Whittington et al 2018) recently showed that place cells tend to remap in new environments to positions that align with a firing field of the grid cell. They had a different interpretation of that result, but it seems potentially consistent with your model as well.

In their account, LEC codes for objects, MEC for structure (associations or links) and the hippocampus codes for the conjunction. For us, the hippocampus codes for the objects or concepts, and mEC monitors the hippocampal place or concept cells. LEC, on the other hand, likely plays a role in object coding, but we do not explicitly assign a role for them in our account. Theirs is a different account and we don't discuss remapping effects, but we are considering how this may fit in future work.

We thank both reviewers for their helpful comments and the opportunity to allow us to clarify our position and address their concerns.

Reviewers' Comments:

Reviewer #2:

Remarks to the Author:

The authors have provided useful responses to my previous points. I have in fact misunderstood one core aspect of the model and apologise if some of my questions reflected this misunderstanding. The authors' responses and changes in the text have made it clearer. This and their other responses have improved the manuscript.

I think the authors' ideas regarding a link between place and grid cells and a broader learning mechanism reflect an interesting and important approach. I still have number of major points that I believe the authors should address however. While the model may be a "high-level" model, it is important that predictions are clarified, and mismatches or missing data are honestly discussed.

It is still unclear to me how exactly place cells relate to the model. The authors emphasise that there "need not be a one-to-one mapping" between concepts and place cells. But the paper remains fuzzy about this point throughout the manuscript (e.g., "clusters function as place cells" line 120, "so called place and grid cells emerge", abstract). So, what are place cells in the model?

This has implications for the relation between place and grid cells according to the model. The error monitoring mechanism should have peaks exactly where existing concepts have their centers, correct? Even if there is no direct one-to-one mapping of clusters to place cells, would there not still be some implications regarding the link between the distribution of place fields and grid cell's firing fields?

Is the model able to explain any effect of grid cells on place cells? While the evidence, as the authors state, shows evidence for a stronger evidence of place on grid cells, it is not true that the reverse effect is entirely absent (see also, Brandon et al. 2014, Neuron, in addition to the cited work).

In my opinion, the authors somewhat overstate novelty. The possibility that processes in the hippocampal-entorhinal system reflect a domain general mechanism has been discussed much recently, based foremost on the Stachenfeld paper (Nature Neuro). These accounts also make clear that the term "cognitive map" has not been tied to the idea that concepts are grounded/based on spatial processes, but rather that both relate to a more general mechanism. From Behrens et al., 2018, Neuron: "map-like representations observed in a spatial context may be an instance of general coding mechanisms capable of organizing knowledge of all kinds"

It is emphasised that previous evidence linked the proposed concept learning mechanism to the hippocampus. Does the cited work provide evidence for the existence of an error monitoring mechanism specifically, or only for concept formation more generally? (looking through the cited evidence, I could only find more general evidence). Given the importance of the error monitoring mechanism in the current approach, it is important to clarify whether there is direct evidence for it or not.

Timescales/learning: The proposed developmental story provided in the response mismatches the author's characterisation of the time courses in the paper, where clearly a relation to learning is made. I still believe questions remain regarding the timescales and order of place and grid cell emergence when animals are exposed to new environments.

Reviewer #3:

Remarks to the Author:

First, I want to thank the authors for their detailed and thoughtful responses.

Major:

A difficulty that remains in evaluating this paper is that it's hard to pin down the exact message. A lot of this response was that we shouldn't take any individual specific predictions of the model too seriously -- it's supposed to be a high level model -- but I believe this high level point **has** been argued before (that grid cells emerge from task statistics -- e.g. dordek et al, stachenfeld et al, banino et al, behrens et al, self-organizing maps models). The edge this paper has on those is that its details are better (more hexagonal grids in square environments with clustering constraints applied to the statistics-extracting process). So in some sense its contributions are, I believe, actually a bit more low-level -- the reason this prior work did not make these arguments convincingly to the authors' satisfaction is because they failed for low-level reasons.

I think I would conclude that the authors model lets us make a better case for this argument than has been made before. It is also a timely response to the recent articulation of the idea that hippocampus is in some sense intrinsically spatial and achieves generality by applying spatial logic to non-spatial problems (eg Bellmund et al).

Perhaps some high level predictions of the model at the intended level of inquiry would be helpful for clarifying.

Also, I still feel that a major theoretical link with prior work is missing. The authors add citations about circle packing math, but not citations about how this has been explored already in the grid cell context. I think for a lot of people, the circle packing argument (grid cells are gridlike because they circle packing) implies a converse (if/when hippocampus is encoding something that is not the plane, they won't be gridlike and/or the optimal packing structure for that problem would be present). I suspect this is some of the confusion behind the port-drinking reviewer #1's review.

Minor:

Not sure why clustering is the superceding concept here -- it seems like "extracting task statistics" or something more general is going on. After all, clustering is a somewhat unnatural way to organize a continuous state space.

Just because MTL does some error learning doesn't mean grid cells are specifically monitoring errors -- I think that is still a large conceptual leap there. It also seems like a bit of an afterthought relative to the main push of the paper (that grid cells come out of a statistical learning approach).

Seems a little unfair to claim that this model has fewer constraints/assumptions/parameters -- e.g. non-negativity is built in because it is clustering. I guess you could say this paper concludes that the constraints that make grid cells are some of the same constraints that characterize clustering, which is actually kind of neat.

A bit confused about how to maintain agnosticism to whether LEC or MEC is encoding certain variables (e.g. objects) when the model seems to suggest space and objects are processed in the same way -- doesn't that imply MEC grid cells are doing it? Or are you saying LEC and MEC are applying a similar computation to different inputs?

Re: The focus of our model is not on development, but on how the grid-like pattern could form over a

learning episode... We note that these are fair questions but are also very fine-grained aspects of a broad new theory we are offering.

Maybe it should go to the supplement.

We thank both reviewers for their thoughtful and constructive comments. We will first address the shared concerns of both reviewers here. We found that a number of concerns revolved around clarifying our key message and unique contribution.

Our basic contribution is a novel non-spatial account of place and grid cells based on clustering models of concept learning that enables a broad link across areas. In terms of David Marr's levels of analysis (Marr, 1982), our model belongs to the algorithmic level, which lies between the computational and implementation levels. Algorithmic models are concerned with the processes and representations used to carry out tasks. Models at the computational level are underspecified with regards to mechanism, instead focusing on the abstract problem description. Implementation-level models specify detailed mechanism, but are often far removed from cognitive aspects and can be hard to interpret without linking back to the computations or algorithms involved. In contrast, algorithmic models are well placed to bridge across levels, being specific on process and representation but abstracted and simple enough to readily link to the neural implementation (for a comprehensive review, see Love, 2015, *Top in Cogn Sci*). Algorithmic models can be used as a lens on implementation-level data, such as brain imaging (e.g. Davis et al., 2012; *JEP:LMC*; Mack, Love, & Preston, 2016; *PNAS*). Specifically, internal representations of these models (e.g. pattern of cluster activations) can be linked to brain activity and show a functional mapping (in contrast to a one-to-one mapping) to brain activity (e.g. using representational similarity analysis; Mack et al., 2016, *PNAS*; for a review, see Turner et al., 2017, *J. Math. Psychol.*), in contrast to a low-level account such as a biologically-inspired spiking neural network that predicts spike timing in a specific context. We make clear in the Discussion how models at different levels of analysis make complementary and mutually constraining contributions to understanding the overall system.

Algorithmic models can bridge theories by linking algorithm to implementation. A body of work across cognitive neuroscience have utilized this approach successfully (e.g. concept learning: Love, 2015; *Top Cogn Sci*.; perceptual decision making: Summerfield & Tsetsos, 2012, *Front Neurosci*; reinforcement learning: Gläscher & O'Doherty, 2010, *WIREs Cogn Sci*). In this contribution, we go one step further and provide an algorithmic model that links across **two** different computational accounts of task descriptions – spatial tasks and concept learning tasks. We propose that these two task domains share a common algorithm and implementation. This aspect of our contribution is unique and distinguishes us from other popular accounts of place and grid cells (e.g. Stachenfeld et al., 2017, *Nature Neuroscience*; Baram et al., 2018, *bioRxiv*) that we also view as primarily algorithmic level accounts (see the list below for three aspects of our account that distinguish it from the aforementioned accounts).

To clarify, our account is 'higher-level' in terms of its motivation (from cognitive processes linking to neural processes; top-down) and its level (algorithmic). This is in contrast with 'lower-level' place and grid cell models, such as those that try to explain specific firing patterns and propose mechanisms that can generate these firing patterns (e.g. spiking models; Lengyel et al., 03, *Hippocampus*; boundary-vector model of place cells; Barry et al., 2006; *Review in the Neurosciences*; oscillatory interference model of grid cells; Burgess et al. 2007; *Hippocampus*).

In our account, an abstract clustering mechanism is proposed to organize information to represent the current environment or task (e.g. concept structure), and the internal

representations of the model, when applied to space, map on to place and grid cell-like activity found in the hippocampal entorhinal cortex. Our algorithmic model is well positioned to bridge across level and domains (Love, 2015). In this contribution, we link model processes that have successfully captured behavior and brain representations during concept learning to spatial representations found at the single-cell level (implementation level). Our model works at the same level of description as alternate algorithmic models such as those that propose that the successor representation is computed and used by the hippocampal formation for organizing information (e.g. Stachenfeld et al., 2017, *Nature Neuroscience*; Baram et al., 2018, *bioRxiv*).

To clarify our unique contributions, we present three key aspects of our account and how they differ to other accounts:

1. In our account, cluster representations (represented by place or concept cells) are in the hippocampus which organize the environment or task information at hand, and grid and non-grid spatial cells in medial entorhinal cortex (mEC) monitor place cell activity. Grid cells do not play a representational role, rather they monitor the representations (place cells) in the hippocampus, and play a monitoring-error role as proposed in concept learning models (in contrast to Stachenfeld, Behrens, and Bellmund).
2. There is no special relation between space and concepts – rather, there is a general learning mechanism (in contrast to Bellmund)
3. Our account is an algorithmic-level model taken from an entirely different domain, where we bridge across concept learning model and brain mechanisms with hippocampal-entorhinal brain representations in spatial navigation, suggesting a shared mechanism across domains. Our approach allows us to link across domains where the higher-level computational goal might appear different, but the underlying process (algorithm) might be the same.

We further discuss differences between our account and others in response 3 to reviewer #2, and response 6 to reviewer #3.

We have added this to the discussion to clarify the level of our account (bold is new):

*“Our higher-level account provides a general theoretical framework applicable to a large range of tasks, in contrast to lower-level models of place and grid cells which give specific predictions in spatial contexts but have less explanatory power to generalize across contexts. Our model’s contribution is providing a general mechanism that could be used across domains. **In terms of Marr’s levels of analysis⁴⁹, our model belongs to the algorithmic level. Models at the algorithmic level are well placed to bridge across levels in that they formalize general processes and representations (i.e., the mechanism) in a manner abstract enough to be related to varied neural instantiations⁵⁰. Here, we provided an algorithmic-level model that links across two different computational accounts of task descriptions (spatial and concept tasks), and connects learning mechanisms from concept learning to spatial representations found at the single-cell level. Specifically, we were able to link the model representations to neural***

representations reported in the spatial literature, closely matching a number of empirical observations.” P17

We will now respond to each of the reviewers' comments in detail.

Reviewer #2 (Remarks to the Author):

1. The authors have provided useful responses to my previous points. I have in fact misunderstood one core aspect of the model and apologise if some of my questions reflected this misunderstanding. The authors' responses and changes in the text have made it clearer. This and their other responses have improved the manuscript.

I think the authors' ideas regarding a link between place and grid cells and a broader learning mechanism reflect an interesting and important approach. I still have number of major points that I believe the authors should address however. While the model may be a “high-level” model, it is important that predictions are clarified, and mismatches or missing data are honestly discussed.

It is still unclear to me how exactly place cells relate to the model. The authors emphasise that there “need not be a one-to-one mapping” between concepts and place cells. But the paper remains fuzzy about this point throughout the manuscript (e.g., “clusters function as place cells” line 120, “so called place and grid cells emerge”, abstract). So, what are place cells in the model?

This has implications for the relation between place and grid cells according to the model. The error monitoring mechanism should have peaks exactly where existing concepts have their centers, correct? Even if there is no direct one-to-one mapping of clusters to place cells, would there not still be some implications regarding the link between the distribution of place fields and grid cell's firing fields?

Characterization of place cells and grid cells in the model:

Thank you for this feedback. As detailed below, we now are clearer on how place and grid cells relate to clusters in the model.

A cluster is an abstract unit in the model. This type of model is often applied to learning phenomena, and clusters are recruited to build up the representation of the context at hand and provide some output for the task at hand (e.g. decision, category membership). Clusters update their positions in representational space during learning for a more accurate representation of the task (minimizing error), and new clusters are recruited with especially surprising errors. In a spatial environment, clusters make up the model's internal representation of physical locations, like our mental representation of the environment. The hippocampus (HPC) represents each location (or concept) in the hippocampus with a place (or concept) cell. But there is likely to be more than one cell per location (or concept) in the HPC. It is worth having more than one cell that represents each relevant concept, and this appears to be true in the

hippocampus (Quiari Quiroga, 2008, *TiCS*; Sparse but not ‘Grandmother-cell’ coding in the medial temporal lobe). In this way, there is a many-to-one functional mapping where several place cells fire at the same location, and this population of cells represents the abstract ‘cluster’ that represents the location. Clusters representations functionally map onto a neural population (e.g. populations of similarly-tuned cells), which allow us to make predictions about neural populations in different environments.

This explanation should also clarify the link between clusters (place cells) and the error monitoring mechanism (mEC grid / spatial cells). As noted by the reviewer, the error monitoring mechanism will have peaks where the concepts have their centers. The ‘clusters’ are abstract, but can be implemented by a population of place cells. There can be more than one place cell representing a cluster, as there are likely to be multiple (mEC grid/spatial) cells that monitor different populations of place cells.

There are interesting open questions on these functional mappings, and future research, including models that incorporate cells into their models, may provide new hints for how information is organized in the HPC-EC circuit (see “Limitations of the model / future work” below).

We have added these points to the manuscript (bold is new):

*“It is important to note that clusters are abstract entities in the model, and there need not be a one-to-one mapping to single concept or place cell (e.g. **a cluster can be represented by a group of place cells with similar tuning (c.f.²²) – a functional mapping of multiple place cells to one cluster, and the place cell population to the whole cluster representation**).” P5*

*“In the spatial case, the clusters **function in a similar way to a population of place cells that code for (i.e., discriminate) locations**.” P7*

And the abstract:

*“so called place and grid cell-like **representations emerge**.”*

Predictions:

Our account is based on an algorithmic clustering model for concept learning that aims to characterize learning mechanisms, and since this characterization is at a certain level of abstraction, the main predictions are higher level, such as changes in the task environment will lead to changes in the cluster representation. Several lower-level predictions made by the model are presented in the paper, in which the cluster representations show a functional mapping to observed neural activity patterns – high grid scores in both circular and square environments, grid score is higher in circles than squares, and a reduction in the grid score in a trapezoid environment. We can further speculate on more specific low-level predictions, given how place and grid cells are related in the model.

One simple prediction is that the mapping from place to grid cells within a context should be predictable. An mEC grid or spatial cell is assumed to receive input from multiple place cells, and that mEC cell should have fields in the same location as the

place cells it receives input from. The prediction is that if we can identify these populations and connections, they should correspond in a predictable fashion. Following from this, if place cells that represent a certain location are inactivated, the corresponding fields of the grid cells that monitor those place cells should also disappear. A strict test would require that all (or at least a majority of) place cells that represent one location (a cluster in the model) are inactivated, then all mEC cells should also lose those fields. This may occur after a short delay, since there may be some residual activity or plasticity in the mEC cell. One interesting prediction is that the fields may start reorganizing to pack the space – predicting place cells might move their fields and grid cells too. There is some evidence showing that the fields of grid cells do reorganize with HPC inactivation (Bonnieve et al., 2013, *Nature Neuroscience*), but this has not been quantified, and concurrent recordings with specific (e.g. optogenetic) manipulation would be required for a better test of these predictions.

Another prediction is related to the higher-level aspect of the model that mEC cells play the same function in spatial and concept cases: monitoring error for updating cluster representations in HPC. One prediction for the error-monitoring mechanism (Love et al., 2004), is that if error is high early in learning such that there is no place field (cluster location) in that region (detected by grid cells monitoring the place cells – low firing rate in grid cells at that location), then either the best matching (closest) cluster (representation in HPC) should update its location to move towards that location in representational space. If the error is sufficiently high, a cluster – and so place field – is created in that location. New clusters will also affect the arrangement of other clusters, in which receptive fields nearby could shift and change their response properties. Relatedly, the model predicts that inactivation of mEC should make learning or updating clusters (new locations or concepts) slower, since the error monitoring mechanism is disrupted. This behavioural effect would be linked to slower appearance of place cells in new environments or new learning. Recent evidence suggests place and grid cells both move towards goals or rewards (Dupret et al., 2010; *Nature Neuroscience*; Boccara et al., 2019, *Science*; Butler et al., 2019, *Science*), and and there seems to be a greater number of place cells recruited near goal locations (e.g. Hok et al. 2007, *JNeuro*, Hollup et al., 2001, *JNeuro*) consistent with more clusters recruited at locations of importance.

Finally, our model predicts that both grid and non-grid spatial cells should perform the same function. Recent work from the spatial domain has already shown that non-grid spatial cells in mEC contain as much information as grid cell and could serve similar functions (Diehl et al., 2017, *Neuron*).

These predictions are difficult (but not impossible) to test since they require recording of a large number of cells in both areas, and ideally performing both spatial and abstract tasks. With technology for concurrent recording of many neurons across brain areas improving rapidly (e.g. neuropixels), this may be possible in the near future.

We have added the predictions related to the monitoring-error mechanism into the Discussion, since it is one of the more novel predictions of our account:

“Our account made several predictions that matched empirical data, where changes in environmental geometry lead to specific changes in the cluster representation. One

novel prediction of our model is that when error is high early in learning for a particular location, mEC cells should show a low firing rate and that best matching place cells should update their tunings to more strongly respond at that location (i.e., cluster updating). Updating a cluster (or recruiting a new cluster) should result in adjustment to the tuning of neighboring clusters, leading to a cascade of changes across place cells. When error is low, this signifies a good match between the environment and one's current knowledge (cluster representation) and experience, and little or no update is necessary. Inactivating the mEC should disrupt the error signal, which should disrupt learning in new environments. Recent evidence suggests place and grid cells both move towards goals or rewards⁵⁷⁻⁵⁹, and there seems to be a greater number of place cells recruited near goal locations^{60,61} consistent with more clusters recruited at locations near the goal. Finally, our model predicts that both grid and non-grid spatial cells should perform the same function, consistent with recent work in the spatial domain which showed that non-grid spatial cells in mEC contain as much spatial information as grid cells and could serve similar functions⁶². P19

Limitations of the model / future work:

Like any model, ours has several limitations. Many of our model's limitations mirror its strengths, namely that our account is simple, straightforward, and posed at the algorithmic level. Accordingly, this higher-level account invites a complementary lower-level account to link the processes and representations we propose to the cellular level.

Clusters in our model are abstract units that together make up the model's internal representations. They do not correspond to individual cells, but they functionally map onto neural representations. This functional mapping could be implemented in the brain, for example, as populations of similarly-tuned cells. But this also means that the model does not make out-of-the-box detailed low-level predictions such as spike timing or specific synaptic changes with learning. It is possible to develop a related model to have cells and test how this interacts with the higher-level representations we propose. Such a model would be more complex, descriptive, and could produce more realistic predictions for spike timing or changes in plasticity under different learning scenarios. Of course, such a model would also make it harder to appreciate the basic linkages we highlight across spatial and non-spatial tasks.

There are some open questions as to how place cell remapping works in our account. For us, place cells do not specifically code for locations, but rather a set of features which could include spatial dimensions. Therefore, it can also treat visual cues as a feature in a spatial environment, such as a place cell fires at a location in a particular environment with a perceptual cue (e.g. a visual cue on a wall). The HPC then can encode the two-dimensional spatial environment in a specific context in a higher-dimensional space. Therefore, place cell remapping can be explained by the HPC population encoding a different context, where there is global shift in the HPC cell population's representation, and place cells change their tuning to properties of the other environment. This is consistent with our account, where the HPC representation consists of all relevant features of the current task or environment, as well as with a body of memory work (context bindings, conjunctions, episodic memory), which our model is highly related to (Mack, Love, & Preston, 2017, *Neuroscience Letters*). This paper is the first step linking disparate literatures using our high-level account.

Currently, our model does not specifically formalize the way this occurs to make specific predictions for which place cells would remap to different ones. However, we hope to combine our model with ideas in memory and learning to investigate models that can explain place cell remapping and make new predictions about remapping in new environments – spatial or abstract. Future work that incorporates more low-level mechanisms such as those mentioned above might help understand how remapping mechanisms relate to the HPC's general representational coding properties.

We have highlighted this and proposed new directions to link our model with lower-level mechanisms in the Discussion:

“The primary strength of our account, namely that it offers an algorithmic account of spatial and concept learning tasks, serves to highlight the need for complementary lower-level accounts. There are various open questions such as how place cell remapping occurs across contexts and partial remapping effects with disruption to mEC^{63,64}. Our hope is that our model can eventually link to lower-level models that incorporate biological details such as spiking neurons and incorporate knowledge from memory research that can explain more empirical findings and provide new insights to these questions. Accounts are needed at multiple levels of analysis. We view our model as intermediary (at the algorithmic level) and aim for it to serve as a bridge between the goal of the computation and its implementation. Our model can serve as a guide for how operations such as cluster updating are physically realized.” P20

2. Is the model able to explain any effect of grid cells on place cells? While the evidence, as the authors state, shows evidence for a stronger evidence of place on grid cells, it is not true that the reverse effect is entirely absent (see also, Brandon et al. 2014, Neuron, in addition to the cited work).

Our model holds that mEC grid and spatial cells arise from monitoring place cell activity in the HPC, and therefore contain information about how far the current location is from a population of place cells. Although there is no representational role for the mEC cells, it has an important role in learning – it can inform the HPC about cluster match error in a more global sense, in contrast to individual HPC cells. Specifically, the mEC cells have information on multiple place cells' field locations and therefore the closest place field to the agent, informing which cluster should update its position. If the error is surprisingly high, a new cluster might be recruited. Therefore, as suggested above in the prediction above, the model would predict that mEC lesions would impair learning, especially for surprising or novel examples, which normally would contain the most information when learning. In addition, there would be slower place cell recruitment during new learning or in new environments.

This prediction can be tested during new learning, but it is less clear what will happen after the task or environment is well learnt. For example, partial remapping of place cells after the inactivation of mEC or related areas (e.g. in Brandon et al., 2014, Rueckemann et al., 2016, *Hipp*) occurs after the animals have learnt about the spatial properties of the environment. Future models that take into account mechanisms of remapping may shed light on how these effects arise.

We have also cited Brandon et al. (2014, *Neuron*) and Rueckemann et al. (2016, *Hipp*) in the manuscript when discussing the effect of disrupting mEC signals on place cells.

3. In my opinion, the authors somewhat overstate novelty. The possibility that processes in the hippocampal-entorhinal system reflect a domain general mechanism has been discussed much recently, based foremost on the Stachenfeld paper (*Nature Neuro*). These accounts also make clear that the term “cognitive map” has not been tied to the idea that concepts are grounded/based on spatial processes, but rather that both relate to a more general mechanism. From Behrens et al., 2018, *Neuron*: “ map-like representations observed in a spatial context may be an instance of general coding mechanisms capable of organizing knowledge of all kinds”

At the beginning of this letter, we list three distinguishing aspects of our account in relation to competing accounts. One unique proposal is that this particular learning process across concept learning task and spatial tasks are shared – that the hippocampal formation may use a clustering algorithm to organize information. The novel contribution is an algorithmic-level account that bridges across two different computational accounts of task descriptions and domains – spatial and concept tasks – and links learning mechanisms and brain representations from concept learning to spatial neural representations found at the single-cell level.

In reference to the ‘cognitive map’ not being always tied to the spatial context, we have edited the abstract and introduction (removing the part that suggests ‘cognitive map’ must mean a spatial cognitive map). We have also added this and appropriate citations on related non-spatial accounts (Stachenfeld, Behrens) to the introduction:

“Such a learning system would be tasked with learning all relevant concepts, including those tied to physical space (also see^{14,15}, and Discussion).” P4

We take this opportunity to clarify our account and compare it with related views that consider the hippocampal-entorhinal place-grid cell circuit role in organizing non-spatial information. In our account, cluster representations which organize the environment or task information at hand reside in the hippocampus, and the medial entorhinal cortex (mEC) monitors place cell activity in the HPC, monitoring error of cluster match. Notably, grid cells in mEC are only a subset of these error monitoring cells – both grid and non-grid spatial cells in mEC can perform this function, and the “grid-ness” of a subset of these cells is a result of the environment or space. All other accounts specifically consider grid cells (but not non-grid spatial cells in mEC) a key representational unit of the cognitive map (Stachenfeld: place cells encode predictions of future states, grid cells are low-dimensional projection of the place cells, Behrens: place cells are conjunctions of objects and structure, grid cells representing learnt structure; Bellmund; grid cell represents spatial structure which directly maps onto abstract/concept structure).

The way in which abstract structure is represented in our account is markedly different. Stachenfeld et al. (2017) suggested that place cells encode predictions of future states, and grid cells encode a low-dimensional decomposition of this hippocampal “predictive

map”. This account proposes that the hippocampal-entorhinal circuit computes the successor representation – an algorithmic model – to organize information (see also Baram et al., 2018, *bioRxiv*). In both Stachenfeld’s and Behrens’ accounts (Baram et al., 2018), they consider the grid cell representation as a representation of the structure of future or possible states. In contrast, in our account grid cells monitors error (of place cells), which contains information about structure of place cells (as existing clusters) – and it can inform the HPC system to update the cluster representation by moving the closest existing cluster or create a new cluster, which updates the representational structure of the environment or task in HPC.

Behrens and colleagues (e.g. Whittington et al., 2018, *bioRxiv*), propose that the hippocampal-entorhinal circuit learns and represent structural knowledge useful for generalization. They suggest that objects are represented in lateral entorhinal cortex (IEC), structure is represented in mEC, and the HPC encodes conjunctions of the two. The structure represented by mEC can be used to generalize to different contexts, assuming there is shared structure. We do not assume this object/structure distinction in the IEC/mEC in their contribution to HPC representations (also see response #6 to reviewer 3 below). In contrast, we think that the representational content and structure is constructed in the HPC, and any generalization to new instances from existing structure is in the HPC (as suggested in clustering models of concept learning). In our account, the structure learnt in the hippocampus is flexible in that it can consider any relevant dimension (object feature, spatial, etc.), whereas their account seems committed to the object-structure distinction which is combined in the hippocampus. For example, a concept may be defined by several object features, and the mEC represents the structure (feature combinations) and IEC the features, rather than objects. However, as far as we know, the IEC preferentially processes objects, and it is unclear how their model accounts for this. Furthermore, we think that the task and context-relevant HPC representations come in part from prefrontal cortex (task/feature representation; dorsolateral and medial prefrontal cortex), rather than purely combining EC inputs.

Whereas our account holds that place and grid cells emerge from a general learning system, Bellmund and colleagues suggest that place and grid cells play a key role in mapping the dimensions of cognitive spaces in cognitive tasks. The proposal suggests that there is a one-to-one mapping from neural representations of physical space to abstract space, and that place and grid cells provide the metric or distance code for abstract spaces. This seems to suggest the system can only code for abstract tasks that can be mapped to spatial dimensions, and we argue that, in many cases, there is no simple mapping from task to physical space. For example, when a context requires a significant degree of selective attention to stimulus features or task variables, the representational space can be warped to a different, more effective representation of the context at hand, which cannot be simply mapped onto the two-dimensional spatial case.

We have added the following to the Discussion:

“Other accounts hold that grid cells are key representational units in the cognitive map. For example, Stachenfeld and colleagues¹⁴ suggested that place cells encode predictions of future states, and grid cells encode a low-dimensional decomposition of this hippocampal “predictive map” that may be useful for stabilizing the map and

representing sub-goals. In contrast, we suggest that place cells (clusters) are the key representational units which encode locations in representational space and its structure, whereas grid cells monitor place cell activity. Behrens and colleagues^{15,45} proposed that the hippocampal-entorhinal circuit learns and represents structural knowledge useful for generalization. This account assumes objects are represented in lateral EC (IEC), structure is represented in mEC, and the hippocampus encodes conjunctions of the two. The learnt structural information in mEC can be used to generalize to different contexts with shared structure. In our account, conceptual knowledge and its structure is represented in the hippocampus, and any generalization to new instances from existing structure is from hippocampal representations (as generalization is performed in clustering models of concept learning). In contrast to the view that hippocampal representations arise from interactions between mEC and IEC, we argue for a central role of prefrontal cortex (representation of the task or relevant features) for shaping hippocampal representations, in combination with sensory inputs arriving via entorhinal and perirhinal cortex, and from anterior inferior temporal cortex to prefrontal cortex^{46,47} to the hippocampus.

Whereas our account holds that place and grid cells emerge from a general learning system, Bellmund and colleagues suggest that the population code of place and grid cells play a role in mapping the dimensions of cognitive spaces in cognitive tasks, and that spatial navigation could serve as a model system to understand cognitive spaces¹⁰ (also see³). Although there are commonalities, their proposal suggests that place and grid cells provide or a 'metric' or distance code for abstract spaces, and that there is a straightforward mapping from neural representations of physical space to abstract space. In our view, when the context involves a significant degree of selective attention to stimulus features or task variables, the representational space can be warped to a different, more effective representation of the context at hand (e.g. reducing dimensionality by attending to the task-relevant dimensions⁴⁸), which does not simply map onto the two-dimensional spatial case." P15

4. It is emphasised that previous evidence linked the proposed concept learning mechanism to the hippocampus. Does the cited work provide evidence for the existence of an error monitoring mechanism specifically, or only for concept formation more generally? (looking through the cited evidence, I could only find more general evidence). Given the importance of the error monitoring mechanism in the current approach, it is important to clarify whether there is direct evidence for it or not.

Current evidence for an error-monitoring signal has been found during concept learning tasks in fMRI (Davis et al., 2012; *JEP:LMC*; the brain signal parametrically varies with the monitoring error parameter estimated in the models): "The entropy measure indexes the degree to which the model is uncertain about the cluster assignment for a stimulus ... activation that was correlated with the entropy measure was observed in both the anterior hippocampus ... striatum ... right posterior parahippocampal cortex... left entorhinal cortex, and left perirhinal cortex." P831.

It is notable that the error-monitoring signal was not only found in mEC, and we also do not expect this signal to only be useful in this brain region. It is a signal that can be

used by any other brain region, and can depend on the task context. For instance, in that study, stimuli were objects, which preferentially engage the perirhinal cortex. The entropy signal in both entorhinal and perirhinal cortex is interesting. There is some recent evidence that hints that spatial cells like those found in the hippocampal formation can be found elsewhere in the brain (e.g. LGN; Hok et al., 2018, *bioRxiv*; somatosensory cortex; Long & Zhang, 2018, *bioRxiv*), though this early evidence must be confirmed.

Given the abundance of neurophysiological work in the hippocampal formation and known link between place and grid cells, we came to focus on this particular system, where the clear circumscribed place cell fields and spatial cells with multiple fields in the mEC system is consistent with a clustering and monitoring error account (mEC cells monitoring error, or cluster match, of HPC cells) used in clustering models concept learning – which is an interesting parallel. The error-monitoring signal is theoretically interesting in concept learning, but has not been a relevant theoretical construct for the spatial processing field. It is not too surprising, then, that there is no direct evidence from animal neurophysiology. Further research is required to test this idea, as discussed in the predictions above.

5. Timescales/learning: The proposed developmental story provided in the response mismatches the author's characterisation of the time courses in the paper, where clearly a relation to learning is made. I still believe questions remain regarding the timescales and order of place and grid cell emergence when animals are exposed to new environments.

We may not have been clear on the explanation of our interpretation of the model and how it links to development. First, we are making the link between 'development' and 'learning' in the model in the sense that during development, rats are learning about spatial environments over time, whilst the cells are developing. The developmental story relates to a certain interpretation of the model's grid cell-like representation. That is, clusters are like place cells, and grid cells monitor the place cells – and that before the clusters have learnt the structure of the environment, the grid cells are not grid-like yet (the clusters have not been organized yet), like mEC cells in baby rodents.

We have added this in the Discussion for clarification:

*"This is consistent with developmental work^{52,53}, where place cells appear in baby rats very early in life, and grid cells develop shortly after **as they explore and learn about spatial environments during normal development.**" P18*

New environments:

Predictions in new environments, especially for humans, are difficult because of prior knowledge. When modeling concept learning or learning abstract task structure, a novel task context is often considered entirely new and a new set of clusters are recruited. However, we have rich priors about task as well as spatial structure and often reuse this prior knowledge. This is an active area of research in the concept learning field, and we are considering ways to incorporate these ideas in our future models. There is a good deal of work on structure learning and generalization to other

similar contexts (e.g. Harlow, 1949; Whittington et al., 2018, *bioRxiv*; Luyckx et al., 2019, *eLife*; also c.f. recent deep learning work), and we plan to examine this in concept learning tasks with different task structures. One of our predictions is that in entirely new environments, that place cells should appear before grid cells (but this could be very quick due to priors), and that grid cells should become more grid-like over time. This latter point seems to be the case (e.g. Sun & Giocomo, 2018, *SfN abstract 604.25*), but may also be due to more sampling over longer periods of testing. Several forthcoming studies from this group and others may illuminate us on this matter soon.

Reviewer #3 (Remarks to the Author):

First, I want to thank the authors for their detailed and thoughtful responses.

Major:

1. A difficulty that remains in evaluating this paper is that it's hard to pin down the exact message. A lot of this response was that we shouldn't take any individual specific predictions of the model too seriously -- it's supposed to be a high level model -- but i believe this high level point *has* been argued before (that grid cells emerge from task statistics -- e.g. dordek et al, stachenfeld et al, banino et al, behrens et al, self-organizing maps models). The edge this paper has on those is that its details are better (more hexagonal grids in square environments with clustering constraints applied to the statistics-extracting process). So in some sense its contributions are, I believe, actually a bit more low-level -- the reason this prior work did not make these arguments convincingly to the authors' satisfaction is because they failed for low-level reasons.

I think I would conclude that the authors model lets us make a better case for this argument than has been made before. It is also a timely response to the recent articulation of the idea that hippocampus is in some sense intrinsically spatial and achieves generality by applying spatial logic to non-spatial problems (eg Bellmund et al).

Perhaps some high level predictions of the model at the intended level of inquiry would be helpful for clarifying.

We thank the reviewer for pushing us to better clarify our contribution. At the beginning of this letter, we listed three key contributions and how they differ from existing work. One key message is that clustering models of concept learning used to model categorization behavior (with a correspondence with hippocampal representations) can also lead to grid cell-like activity. This links a body of concept learning work – how clustering models account for concept formation and inference and generalisation – with spatial cells, suggesting they might share a common neural mechanism (also see response #1 to reviewer 2 clarifying how abstract clusters link with place cells and grid cells, and the “Predictions” section for some new predictions of the model). We also discuss specific differences between our account and other recent accounts (Stachenfeld, Behrens, Bellmund) in response 3 to reviewer 2 above, and in response 6 below.

2. Also, I still feel that a major theoretical link with prior work is missing. The authors add citations about circle packing math, but not citations about how this has been explored already in the grid cell context. I think for a lot of people, the circle packing argument (grid cells are gridlike because they circle packing) implies a converse (if/when hippocampus is encoding something that is not the plane, they won't be gridlike and/or the optimal packing structure for that problem would be present). I suspect this is some of the confusion behind the port-drinking reviewer #1's review.

Thank you for highlighting this relevant research. We have now cited the following papers: Kropff & Treves, 2008, *Hippocampus*; Mathis et al., 2015, *eLife*, Wei et al., 2015, *eLife*, Dordek et al., 2016, *eLife*. Please feel free to alert us if you had some other specific paper in mind.

We have also added this to the discussion:

“Other work have modeled or analyzed mathematical properties of the grid code (e.g.^{31,33}), but also do not account for variability in the grid score in mEC cells.” P17

Minor:

3. Not sure why clustering is the superceding concept here -- it seems like “extracting task statistics” or something more general is going on. After all, clustering is a somewhat unnatural way to organize a continuous state space.

Clustering models of concept learning suggest that clustering is the process used in concept formation. It is also an algorithmic model (cf. Love et al., 2004; Sanborn et al., 2010, *Psych Review*) that aims to capture the processes the brain uses, rather than a purely statistical model that simply describes the data. For us, modelling the learning process is key, and a statistical model or description would not be sufficient.

Clustering is a flexible process and can be used to extract statistics about the task or spatial environment. In the field of statistics, clustering algorithms are commonly used in continuous spaces, such as density estimation, which is ideal for capturing statistics in a continuous state space. One can see the clusters as the relevant, incrementally learned, bases to the space, much like how principal components in PCA provide a bases for continuous spaces. Finally, spatial environments are not always perfectly smooth – there are discontinuities such as doors, inaccessible paths (e.g. a hole in the ground), or cliffs.

In our view, clustering is a useful way to organize continuous state space to separate it into useful parts, making it more compact and efficient. Humans categorize things, use landmarks, and mark ‘areas’ of interest. Clustering models successfully capture a large variety of human categorization behavior, and here, we show an interesting parallel with the place/grid cell representation of physical space, suggesting that a common mechanism might be involved.

4. Just because MTL does some error learning doesn't mean grid cells are specifically monitoring errors -- I think that is still a large conceptual leap there. It also seems like a bit of an afterthought relative to the main push of the paper (that grid cells come out of a statistical learning approach).

We might have adopted a confusing phrasing here. We specifically refer to error as a cluster match (internal) error, not a task-related error. To clarify, the monitoring-error signal in clustering models of concept learning are not the same as error signals as typically considered in other theories (like prediction errors in reinforcement learning models). The observed mEC grid and spatial cell activation patterns are consistent the error-monitoring signal in the clustering model, where it monitors the activity of place cells and therefore contains information about how far the current location is from the existing cluster/place cells it receives input from. Specifically, the mEC grid and spatial cell activity patterns, with multiple peaks spread out equally, is consistent with this idea.

Furthermore, monitoring error is a key feature in these models, rather than an afterthought (see Anderson, 1991; Love et al., 2004, Psych Review). It was of particular theoretical interest to us that it matches what grid cells and other non-grid spatial cell in mEC show, since it suggests a common mechanism. There is some evidence the regions in the MTL, including mEC, show this signal (Davis et al., 2012; *JEP:LMC*), and future work can test for this signal and its relevance in spatial and non-spatial tasks (also see response 4 to reviewer 2). We have also discussed some predictions related to the monitoring-error hypothesis in response 1 to reviewer 2, under the header “Predictions”.

We have added this to the manuscript for clarity:

“Unlike prediction errors in reinforcement learning models, which are signed differences between prediction and observed reward, these cluster signals are concerned with match to learned internal representations.” P5

5. Seems a little unfair to claim that this model has fewer constraints/assumptions/parameters -- e.g. non-negativity is built in because it is clustering. I guess you could say this paper concludes that the constraints that make grid cells are some of the same constraints that characterize clustering, which is actually kind of neat.

We thank the reviewer for this comment – this is a nice characterisation and we have added it to the discussion:

*“Here, we used a simple model from a high-level perspective based on ideas from concept learning and memory and matched the proportion of grid cells with empirical data, **suggesting that the constraints of the clustering model matches the constraints the brain uses to build these representations.**” P18*

6. A bit confused about how to maintain agnosticism to whether LEC or MEC is encoding certain variables (e.g. objects) when the model seems to suggest space and objects are processed in the same way -- doesn't that imply MEC grid cells are doing it? Or are you saying LEC and MEC are applying a similar computation to different inputs?

Apologies that we were unclear in our previous response on this issue. Indeed, our account suggests that HPC and mEC process any task-relevant variables – including

objects and spatial variables – in the same way. However, we are not tied to the IEC/mEC-object/space distinction. We explain this in full below.

A common distinction is that the mEC preferably processes spatial information and IEC processing object information. This is also what Behrens and colleagues commit to in their account. We discuss this in response 3 to reviewer 2 above, and include the relevant section here:

Behrens and colleagues (e.g. Whittington et al., 2018, *bioRxiv*), propose that the hippocampal-entorhinal circuit learns and represent structural knowledge useful for generalization. They suggest that objects are represented in lateral entorhinal cortex (IEC), structure is represented in mEC, and the HPC encodes conjunctions of the two. The structure represented by mEC can be used to generalize to different contexts, assuming there is shared structure. We do not assume this object/structure distinction in the IEC/mEC in their contribution to HPC representations (also see response #6 to reviewer 3). In contrast, we think that the representational content and structure is constructed in the HPC, and any generalization to new instances from existing structure is in the HPC (as suggested in clustering models of concept learning). In our account, the structure learnt in the hippocampus is flexible in that it can consider any relevant dimension (object feature, spatial, etc.), whereas their account seems to be more committed to the object-structure distinction which is combined in the hippocampus. For example, a concept may be defined by several object features, and the mEC represents the structure (feature combinations) and IEC the features, rather than objects. However, as far as we know, the IEC preferentially processes objects, and it is unclear how they account for this. Furthermore, we think that the task and context-relevant HPC representations come in part from prefrontal cortex (task/feature representation; dorsolateral and medial prefrontal cortex), and in part from sensory inputs. These inputs may come from EC and perirhinal cortex, but also note that in humans, visual inputs from another route of the ventral stream (early visual cortex to anterior inferior temporal cortex) go to (inferior) PFC through the multiple white matter tracts (including the arcuate/superior longitudinal, and inferior occipitofrontal fasciculi; Martino et al., 2011; *Journal of Anatomy*; Catani et al., 2002, *NeuroImage*), which then can send this information to the HPC.

Of course, there are object and spatial cells in these respective areas. One possibility is that both perspectives account for different parts of the system (e.g. separate populations and computations). Alternatively, they could be interlinked (e.g. IEC/mEC relays processed sensory information to the HPC, which is integrated with task-related signals from the PFC, and the HPC sends information back to IEC/mEC which modulates its representations). We look forward to future work with large-scale recordings combined with varied cognitive tasks which may provide new insights on these questions.

7. Re:The focus of our model is not on development, but on how the grid-like pattern could form over a learning episode... We note that these are fair questions but are also very fine-grained aspects of a broad new theory we are offering.

Maybe it should go to the supplement.

We tried to strike a balance here as Reviewer 2 wishes us to include some discussion of this issue. We have not elaborated further on this point though.

Reviewers' Comments:

Reviewer #2:

Remarks to the Author:

Review Mok and Love

I thank the authors for providing a substantial revision and addressing my concerns. In my opinion, many issues are now resolved, and I am convinced this manuscript will make a great contribution to the field. I still have a number of reservations that mainly refer to clarity of the account and its predictions.

Overall contribution: In my opinion the paragraph about Marr's levels is pretty general and does not add anything that would help clarifying the contribution of the model. Most people know about Marr's levels and their strengths, and the present paper does not add anything to these, nor is it designed to make any contribution.

Place-Grid cell relations: I believe this important aspect of the account has become more clear now, but I would encourage the authors to improve Figures 1/2 in a way that makes clear that (a) sampling of feature combinations changes the distribution of *clusters* (b) the distribution of clusters determines for which spatial locations a population of place cells forms receptive fields, but more than one cell could reflect the same cluster (identical/overlapping spatial receptive fields), (c) grid cells reflect the distribution of clusters (and thereby, place cells).

3. Predictions: I think the paper would benefit from a section that comes before the discussion that lists the predictions of the model, as stated by the authors throughout their answers and the paper: The mapping from place to grid cells within a context should be predictable. The specific nature of asymmetric effects of mEC and HPC inactivations onto each other is also predictable. Sampling of the environment and distribution of place fields should interact (I think the reward effects on place cells (2018 paper by Gauthier from Tank lab) and on grid cells (Butler et al, 2019 in Science) seem directly in line with this? There should be effects of HPC and mEC inactivations in *concept learning* in animals. The model would make very specific predictions about how place and grid cells form during learning. Plus, as the authors say, in entirely new environments place cells should appear before grid cells

4. The paper could also benefit from not mingling the issues of learning mechanism and representations as much. A sentence saying that the assumption is that (a) there is a learning mechanism that reduces a specific cost function (which?) and (b) this mech in turn leads, at the end of learning, to place and grid cells, would very helpful.

5. Minor: The sentence in line 71 does not make sense to me: RPEs don't seem unlike the current mechanisms in the sense that is stated here: Values expectations are also internally learned representations, right?

Reviewer #3:

None

We thank the reviewer for their positive comments and final suggestions. Please see our responses below.

Reviewer #2:

1. Overall contribution: In my opinion the paragraph about Marr's levels is pretty general and does not add anything that would help clarifying the contribution of the model. Most people know about Marr's levels and their strengths, and the present paper does not add anything to these, nor is it designed to make any contribution.

We have removed the two sentences about Marr and the algorithmic level - *"In terms of Marr's levels of analysis⁴⁹, our model belongs to the algorithmic level. Models at the algorithmic level are well placed to bridge across levels in that they formalize general processes and representations (i.e., the mechanism) in a manner abstract enough to be related to varied neural instantiations."*

We kept the paragraph to clarify our higher-level aspect of the model, and to distinguish it with lower-level place and grid cell models.

2. Place-Grid cell relations: I believe this important aspect of the account has become more clear now, but I would encourage the authors to improve Figures 1/2 in a way that makes clear that (a) sampling of feature combinations changes the distribution of *clusters* (b) the distribution of clusters determines for which spatial locations a population of place cells forms receptive fields, but more than one cell could reflect the same cluster (identical/overlapping spatial receptive fields), (c) grid cells reflect the distribution of clusters (and thereby, place cells).

We have added to the Fig. 1 legend: *"How the stimulus space is sampled affects how clusters are distributed in the representational space."*

We have added Fig. 2C, illustrating that how the clusters relate to neurons. We have added to the legend: *(C) Clusters determine the receptive fields for a population of place or concept cells, and the cluster-monitoring/error-monitoring mechanism (grid or spatial cells) reflect the distribution of the clusters. Abstract cluster representations are instantiated by multiple cells in the hippocampus and medial entorhinal cortex (mEC) with similar firing fields to represent the same location (or concept) in the case of hippocampal cells or cluster match in the case of mEC cells."*

3. Predictions: I think the paper would benefit from a section that comes before the discussion that lists the predictions of the model, as stated by the authors throughout their answers and the paper:

The mapping from place to grid cells within a context should be predictable. The specific nature of asymmetric effects of mEC and HPC inactivations onto each other is also predictable.

Sampling of the environment and distribution of place fields should interact (I think the reward effects on place cells (2018 paper by Gauthier from Tank lab) and on grid cells (Butler et al, 2019 in Science) seem directly in line with this?)

There should be effects of HPC and mEC inactivations in *concept learning* in animals. The model would make very specific predictions about how place and grid cells form during learning. Plus, as the authors say, in entirely new environments place cells should appear before grid cells.

We have added detail on these predictions in the discussion section (p19-20). Since predictions of the model are not part of the results, it is difficult to place them in a section before the Discussion section as the reviewer suggested. We decided to add these to the predictions section in the discussion. Please note that some of these predictions and citations (sampling the space, and reward/goals - Butler/Boccaro) have been mentioned. The Gauthier/Tank lab paper is more specific to a pool of reward neurons in the hippocampus so may be less relevant here.

“The model also provides further predictions. First, the mapping from place to grid cells within a context should be predictable. An mEC grid or spatial cell is assumed to receive input from multiple place cells in the hippocampus, and that mEC cell should have fields in the same location as the place cells it receives input from (Fig. 2A-B). Therefore, if place cells that represent a certain location are inactivated, the corresponding fields of the mEC cells that monitor those place cells should also disappear. Since an mEC cell may receive inputs from multiple place cells, a strict test would require inactivation of all (or at least a large proportion of) place cells that represent one location (a cluster in the model), predicting all mEC cells should also lose those fields. Future work with large-scale concurrent recordings in multiple brain regions with specific (e.g. optogenetic) manipulation may allow these predictions to be tested.”

“Finally, our model predicts that both grid and non-grid spatial cells should perform the same function, in both concept and spatial tasks.”

4. The paper could also benefit from not mingling the issues of learning mechanism and representations as much. A sentence saying that the assumption is that (a) there is a

learning mechanisms that reduces a specific cost function (which?) and (b) this mech in turn leads, at the end of learning, to place and grid cells, would very helpful.

Added to discussion (bold is new):

*"Here, this same basic account was shown to account for basic spatial navigation phenomenon, including place and grid cell-like response patterns. **Specifically, we showed that a learning mechanism that seeks to minimize error in the task-relevant feature space captures conceptual structure in concept learning tasks and spatial structure in two-dimensional navigation contexts, which lead to place and grid cell-like representations.**"*

We hope this also clarifies the idea that there is a learning mechanism that lead to particular representations in the model, which is what we propose occurs in the brain.

5. Minor: The sentence in line 71 does not make sense to me: RPEs don't seem unlike the current mechanisms in the sense that is stated here: Values expectations are also internally learned representations, right?

We have removed this sentence.